# Configuration and Evaluation of a Global Unstructured Mesh Atmospheric Model (GRIST-A20.9) based on the Variable-Resolution Approach

**Yihui Zhou[1,2], Yi Zhang[3], Jian Li[3], Rucong Yu[3], Zhuang Liu[4]**

[1]State Key Laboratory of Numerical Modeling for Atmospheric Sciences and Geophysical Fluid Dynamics (LASG), Institute of Atmospheric Physics, Chinese Academy of Sciences, Beijing, China

[2]University of Chinese Academy of Sciences, Beijing, China

[3]State Key Laboratory of Severe Weather (LaSW), Chinese Academy of Meteorological Sciences, China Meteorological Administration, Beijing, China

[4]National Supercomputing Center in Wuxi, Jiangsu, China

***Correspondence to***: Yi Zhang (yizhang@cma.cn or yizhang@cma.gov.cn)

**Abstract.** Targeting a long-term effort towards a variable-resolution (VR) global weather and climate model, this study systematically configures and evaluates an unstructured-mesh atmospheric model based on the multiresolution approach. The model performance is examined from dry dynamics to simple and full physics scenarios. In the dry baroclinic wave test, the VR model reproduces comparable fine-scale structures in the refined regions as a fine-resolution quasi-uniform (QU) mesh model. The mesh transition zone does not adversely affect the wave pattern. Regional kinetic energy spectra show that the fine-scale resolving ability improves as the fine resolution increases. Compared to a QU counterpart that has equivalent degrees of freedom, while the VR model tends to increase the global errors, the errors can be reduced when the resolution of the coarse region is increased. The performance over the coarse region is generally close to that of a low-resolution QU counterpart. Two multi-region refinement approaches, the hierarchical and polycentric refinement modes, further validate the model performance under the multiresolution refinement. Activating hyperdiffusion for horizontal velocity is helpful with respect to VR modeling. An idealized tropical cyclone test is further used to examine its ability to resolve fine-scale structures. In the simple physics environment, the VR model can have the tropical cyclone stably pass the transition zone in various configurations. A series of sensitivity tests examines the model performance in a hierarchical refinement mode. The simulations exhibit consistency even when the VR mesh is slightly perturbed by one of the three parameters that control the density function. The tropical cyclone, starting from the 2nd-refinement region and passing through the inner transition zone, gets intensified and covers a smaller area in the refined regions. Such variations are consistent with the behavior that one may observe when uniformly refining the QU mesh. In the full physics environment with a highly variable mesh that reaches sub-10-kilometer resolution, the VR model also produces a reasonable evolution for the tropical cyclone. The explicit diffusion shows its usefulness in terms of suppressing some unrealistic isolated-scale structures that are far away from the initial vortex, and does not adversely affect the physically important object. The fine-scale structure is determined mainly by the fine-resolution area, although the systems may have larger differences before they move into the fine-resolution area. Altogether, this work demonstrates that the multiresolution configuration is a reliable and economic alternative to high-resolution global modeling. The adverse impact due to mesh transition and the coarse region can be well controlled.

## 1. Introduction

Increasing resolution is generally regarded as an effective way to improve global weather and climate modeling (Jung et al. 2012; Wehner et al. 2014; Zhang et al. 2014; Yu et al. 2019). It is apparent that more computational and storage resources are required for higher resolution models. This leads to a major challenge for efficient model development and application. The emergence of the locally refined, variable-resolution (VR) modeling approach offers a complementary route. The term VR is a broad concept. It may be realized with different styles, such as nested regional modeling with multiple grids, abrupt nonconforming mesh division, stretched grids, and the multiresolution approach. The stretched grid (e.g., Hourdin et al. 2006; Harris et al. 2016) and the multiresolution approaches (e.g., Ringler et al. 2011; Guba et al. 2014) are close in terms of their conforming style. They maintain the global modeling configuration while permitting increased resolution for certain regions. The multiresolution approach is usually realized by an unstructured mesh model, such that a more flexible resolution choice can be achieved by considering multiple regions. Such a global-to-regional approach is the VR style that will be investigated in this study.

Numerical weather and climate modeling has shown that the VR approach can preserve the benefits of high-resolution applications for certain regions at a lower computational cost, as the total number of grid points can be greatly reduced (e.g., Sakaguchi et al. 2015; Skamarock et al. 2018; Gettelman et al. 2018). This advantage is especially valuable for high-resolution modeling that may reach the convection-permitting regime. While the global cloud/storm-resolving (a.k.a., convection-permitting/allowing) modeling approach has been widely adopted (e.g., Stevens et al. 2019), it is still expensive and inefficient to frequently run such models for routine model development, research and application. The VR model provides an efficient testbed for evaluating global model configurations and testing scale-aware physics. It offers flexible resolution configurations that may depend on physical interests. When properly formulated, it can be an intermediate and transitional step before establishing global convection-permitting modeling.

While the VR approach has shown some benefits, it may potentially suffer from some problems[1]. The nonuniform mesh, though it can be gradually refined, hardly decreases the global errors (as compared to its uniform counterpart that has the same degrees of freedom) because the truncation error is controlled mainly by the coarse-resolution region (Weller et al. 2009; Ringler et al. 2011); the numerical convergence rate may also be affected (e.g., Düben and Korn 2014). Mesh refinement also tends to create artificial wave distortion and reflection. This issue is more challenging to the staggered finite-volume methods (Ullrich and Jablonowski 2011), which are widely employed in today's weather and climate models due to their cost-effectiveness. At first glance, this seems to pose some disadvantages. Fortunately, the primary motivation of increasing resolution is to accurately resolve meteorologically important fine-scale structures. This implies that the solutions, in particular at high-wavenumbers, change as the resolution increases. As long as one can maximize the model performance over the refined region, and have good control over the adverse impact due to the non-refined regions, the VR approach based on the staggered finite-volume method is extremely promising. This statement is not intended to diminish the importance of pursuing numerical precision, which is one of several important properties when developing model dynamics. It is hoped that a balanced compromise can be achieved, and thus we can take advantage of this promising approach.

Previous numerical studies have investigated the impact of grid refinement on the solution error in shallow-water models (Ringler et al. 2011; Guba et al. 2014), mainly based on single-region refinement. The impact of the width of the grid transition zone and the densification ratio has been emphasized. On the basis of spherical centroid Voronoi tessellation (SCVT), Ringler et al. (2011) demonstrated that the solution error is controlled primarily by the coarse-resolution region, and suggested that this can help to specify the coarse-mesh resolutions by determining what is an acceptable level of accuracy. They also suspected that the width of the transition zone may lead to increased errors. Liu and Yang (2017) suggested that the width of the transition zone may cause smaller additional errors compared to the

---

[1] We only consider the issues related to model dynamics in this study.

increase in the densification ratio (for the tests they examined). Within a unified global model, these problems may potentially reduce the performance in the refined region.

In terms of resolving fine-scale fluid structures, the tropical cyclone is a useful testbed, and has been frequently used to examine resolution sensitivity. The global VR approach has been employed to simulate tropical cyclones for cost-effective climate simulations (Zarzycki and Jablonowski 2014). The VR model can capture smoother cloud patterns and smoother mid-level jet structures across the grid refined region, leading to enhanced tropical cyclone activities, compared to a nested model in which boundary forcing may have an adverse influence (Hashimoto et al. 2015). Based on a VR configuration, the Community Atmosphere Model (CAM) with a spectral element core (Taylor 2011) well maintains tropical cyclones crossing the transition zone without discernable wave reflection (Zarzycki et al. 2014).

While these earlier studies have reported the benefits of VR modeling, a proper utilization of this technique is still challenging and deserves ongoing exploration. In this study, we systematically configure and evaluate the Global-to-Regional Integrated forecast SysTem (GRIST) atmosphere model based on the VR approach. GRIST is a new modeling system developed on an unstructured mesh, independent from existing atmospheric models available in the community. Previous studies have described the model formulation, and evaluated its performance in shallow-water model tests (Zhang 2018; Wang et al. 2019), 3D dry dynamical core (dycore[2] hereafter; Zhang et al. 2019; Z19 hereafter) tests, and multiscale moist-atmosphere tests forced by simple physics (Zhang et al. 2020; Z20 hereafter). These studies considered mainly the quasi-uniform (QU) mesh. The model configuration and performance remain underexplored when local mesh refinement is considered.

In this study, we describe the model configuration for VR modeling. In particular, we will detail the explicit diffusion option, and demonstrate its impact. We then examine the model behavior, to understand its strengths and weaknesses under various mesh-refinement styles. This work is intended to provide a basis for utilizing GRIST-VR for more realistic modeling in future. To achieve these goals, we adopt two idealized initial atmospheric conditions, endorsed by the Dynamical Core Model Intercomparison Project (DCMIP; Ullrich et al. 2017). They drive the model towards some well-expected behaviors, facilitating a basic understanding of VR modeling. The model is forced from zero physics (i.e., pure dynamics) to simple and full physics, so as to represent an increasing degree of complexity.

The remainder of this paper is organized as follows. Section 2 presents the model description and its configuration for VR modeling. Section 3 describes the mesh configuration. Section 4 examines the VR performance in the dry baroclinic wave test. Section 5 investigates the model sensitivity in the tropical cyclone test. Section 6 presents a summary.

**2. Model description**

**2.1 Model framework and dynamics**

The model evaluated here is a frozen version of the GRIST-Atmosphere model. A20.9 denotes the version frozen at September 2020. The major descriptions (dynamical framework and component coupling) are in Z20 and Z19. GRIST is formulated on an unstructured mesh, which permits the use of SCVT (Ringler et al. 2008; Jacobsen et al. 2013) that enables VR modeling. A dry-mass-based generalized vertical coordinate is used. It allows flexible switching between the hydrostatic (HDC) and nonhydrostatic (NDC) cores. The moist-atmospheric model exactly conserves the dry air mass to within machine roundoff. The sink of the moist total energy is limited to a quite small value. The flux-form scalar variables are formulated in a layer-averaged manner. The momentum variables are formulated in their primitive forms. A vertically semi-implicit approach is used for solving the acoustic equations in the NDC, with explicit Eulerian vertical advection. The HDC is fully explicit.

---

[2] In the context of GRIST, dycore specifically refers to the dry part of the governing equations excluding tracer transport; this should be distinguished from the typical usage of dycore in the literature.

The horizontal discretization is formulated on a hexagonal-C grid, that is, using a staggered finite-volume method.
Thuburn et al. (2009) proposed the key construction of the Coriolis term to achieve desirable mimetic properties.
Ringler et al. (2010) formulated a set of spatial operators for the nonlinear shallow-water equations, under the
constraints of integral invariant conservation and compatible vorticity dynamics. This approach has been
used/examined by the Model for Prediction Across Scales (MPAS; Skamarock et al. 2012; Ringler et al. 2013), ICON-
IAP (Icosahedral Nonhydrostatic model at the Institute of Atmospheric Physics, Gassmann 2013) and DYNAMICO
(Dubos et al. 2015). GRIST adopts two variations that differ from that in Ringler et al. (2010). Zhang (2018) extended
a set of high-order upwind/center flux operators (Skamarock and Gassmann 2011) for approximating the edge-based
potential vorticity flux, and demonstrated that both the higher nominal order and implicit upwind damping improve the
simulated vorticity field. A pure third-order upwind formulation is used in GRIST. The other variation is a redefinition
of the kinetic energy term by blending the original primal-cell value with a reconstructed value from the dual cell
(Gassmann 2013). This helps to alleviate the noise associated with the Hollingsworth instability (Hollingsworth et al.
1983) according to earlier QU model tests. A default coefficient of 0.9 is used following Eq. (20) in Z19.
The C-grid is cost-effective in dealing with the flux-divergence and gradient operators, which constitute the major
horizontal computation involved in a full-fledged atmospheric dynamical core. The potentially adverse dispersion issue
due to increasing mesh discontinuity in the VR mode (Ullrich and Jablonowski 2011) can be well controlled by: (i)
using a smooth and gradual mesh transition (e.g., SCVT); and (ii) using a slight amount of explicit diffusion (as will be
discussed). The basic horizontal operators are nominally accurate to the second-order, while the flux operator can be
approximated using higher order extensions. GRIST has several options for the flux operator. Among these, a nominal
fifth-order upwind formulation can generate the smallest numerical errors (see results in Zhang 2018; Wang et al. 2019),
but is not used for our default configuration as it requires three halo layers (the default number is two)[3]. This formulation
is still instrumental in model development, helping to validate that the parallel computing infrastructure (Liu et al. 2020)
is working correctly when using different minimum halo layers. The pure upwind formulation of Skamarock and
Gassmann (2011) is used for the dycore. When combined with a flux limiter, this scheme can be used for tracer transport
(as in Section 5.2). A Two-step Shape-Preserving Advection Scheme (TSPAS; Zhang et al. 2017; Yu 1994) is also a
major option for tracer transport (as in Section 5.1). Two-time-level single/multistage forward-in-time integration is
used for dycore and tracer transport such that dry air mass and tracer mass are coupled in a consistent manner.
Two initial conditions are used in this study. The baroclinic wave (Jablonowski and Williamson 2006; JW06)
examines the adiabatic behaviors in a dry environment. The solution from a high-resolution run can be used as a
reference solution. The idealized tropical cyclone is initialized following Reed and Jablonowski (2011). It is available
from the DCMIP testing scripts. This test does not support a reference solution. As GRIST uses a dry-mass vertical
coordinate, obtaining the moist-atmosphere state requires some special treatment (see Z20 for details). All the
simulations use 30 full vertical levels with a top at ~2.25 hPa, basically identical to the default CAM5 setup (e.g., Reed
and Jablonowski 2012). Both two tests are short-term deterministic tests (i.e., weather forecasting style), helping to
validate the model configuration. Long term climate modeling based on the VR configuration will be reported elsewhere.
Also note that, while some studies have pointed out that the vertical resolution should increase with horizontal
resolution, we keep it unchanged in this study.
**2.2 Model physics**
GRIST provides a general physics–dynamics coupling interface to incorporate various physics packages. A tailored
package can be used as a plugin, and its development can benefit from the broad community resources. One may add a
specific physics scheme to an existing physics package, or create an entirely new physics package as long as it is

---

[3] Using two halo layers may be possible, but would require a more complicated communication rule than the current one, which is undesirable.

compatible with the current workflow and is scientifically reliable. The surface model (e.g., land or a mixed-layer ocean model), though not used in this study, is coupled in a point-to-point style, and can be shared by different physics configurations. Three physics packages are currently available as basis for continuous research and development. These packages are separate in the sense that they have different physics drivers and data structures. For completeness, we describe them in this section.

(i) The DCMIP simple physics package. In this study, the suite of Reed and Jablonowski (2012) is used. It contains a large-scale condensation process, a surface flux scheme, and a boundary layer process. The sea surface temperature is 29℃ globally. These processes are coupled in a time-splitting manner within the package, and the package is coupled to GRIST in a pure operator-splitting approach (ptend_f2_sudden; see Z20).

(ii) A climate physics package adopted from CAM5. This package is not used in this study, but the details can be found in Li et al. (2020). It is currently being tuned for the HDC that targets long-term climate modeling, and can also be used for short-term integration in a weather forecast mode (e.g., Zhang et al. 2015).

(iii) For GRIST-NDC, which is targeted at simulating nonhydrostatic dynamics in a nonhydrostatic regime, a set of parameterization schemes from the Weather Research and Forecast (WRF) model (Powers et al. 2017) has been implemented, and is used in this study. The detailed schemes include: a six-species cloud microphysics scheme (Lin et al. 1983) from WRF version 2.0; the Tiedtke cumulus scheme (Tiedtke 1989) from WRF version 3.7.1; the YSU (Yonsei University) planetary boundary layer scheme (Hong et al. 2006) and a surface scheme from WRF version 2.0; the longwave radiation scheme is the RRTM (Rapid Radiative Transfer Model) from WRF version 2.0 (Mlawer et al. 1997); and the shortwave scheme is a CAM radiation module from WRF version 3.4.1.

The internal coupling of this package is process splitting (see e.g., a review in Gross et al. 2018 for details). All processes start from the dynamics-updated state and send back their respective tendencies to the dynamical model without modifications of the physics state variables. The exception is that microphysics will update the local physics state variables, so the calling sequence still matters. The physics–dynamics coupling uses a hybrid approach that combines the tendency method (ptend_rk) and the operator-splitting approach, as described in Z20. In particular, radiation heating is carried over the internal integration of the dycore as in a tendency method, and other tendencies (microphysics, boundary layer, cumulus) are updated as in a pure operator-splitting approach (ptend_f2_sudden in Z20). We emphasize that this package (including the choice of different schemes and its internal coupling) is still experimental and preliminary. It has not been comprehensively tuned. Ongoing tests, modifications, and additions are required to refine the performance. In this study, *it is only intended to evaluate the particular performance of VR modeling and the role of explicit diffusion in a full physics scenario.*

**2.3 Time-step choices and explicit diffusion**

For a VR model, the time step is theoretically restricted by its fine-resolution region. As emphasized by one reviewer and based on our own experience, some improper VR configurations may lead to higher stability restriction than their equivalent QU counterparts. For example, with regard to the acoustic-mode filter, Klemp et al. (2018) suggested that their original formulation is more problematic in VR applications, and cannot effectively remove the acoustic noise. Uncontrolled acoustic modes may artificially accumulate energy and thus impose a higher stability restriction. For the configurations and tests that we have examined, a suitable configuration of explicit diffusion is mostly relevant to this issue. The details will be given in the following section.

The explicit diffusion tendencies are generated at the largest time interval of each model step (i.e., physics step). They are coupled to model dynamics as in a tendency method (ptend_rk). Z20 showed that the tendency method has a slightly higher stability restriction than the pure operator-splitting approach (ptend_f2_sudden), but it can benefit from the higher accuracy of the time integrator. The tendency method does not require additional data communication. No explicit diffusion is activated for tracer transport in this paper.

**2.3.1 Smagorinsky diffusion**

Flow-dependent Smagorinsky diffusion (Smagorinsky 1963) is used for velocities and potential temperature, following previous QU model tests. It is activated in all experiments, except the baroclinic wave test at the quasi-uniform G6 resolution. This scheme uses a second-order Laplacian operator multiplied by a flow-dependent eddy viscosity. The eddy viscosity is defined at the edge point. The Smagorinsky diffusion is not very scale selective, but its flow-dependent feature makes it selective in terms of where and how much to diffuse (see e.g., Fig. 9 in Gassmann 2013). The diffusion strength is acceptable overall, as evidenced by the sharp gradient in the QU baroclinic wave test.

In the QU mode, a mean length scale is used for calculating the eddy viscosity. For the VR mode, doing so implies a stronger diffusion (than typically required) for the refinement region. This also leads to a higher stability restriction, especially when the refinement ratio becomes large. In this version, the square of this mean length scale is replaced by the local length product of a pair of crossing edges. In the tropical cyclone test (simple physics), this local approach increases the maximum wind magnitude in the eyewall (as compared to the unscaled version in the preprint), because a smaller amount of diffusion is imposed on the refinement area (see Section 5.1). It also reduces the parametric sensitivity to the Smagorinsky coefficient as found in the preprint. When varying the coefficient, the present version generates more consistent solutions, similar to a QU model. This local scaling approach was used for the VR mode but not for the QU mode.[4]

**2.3.2 Hyperdiffusion**

The hyperdiffusion option was not used in the initial preprint. Based on some exploration and experiments, we have found that activating scale-selective fourth-order hyperdiffusion for the horizontal velocity shows demonstrable added value for VR modeling. First, in the baroclinic wave test, in the absence of hyperdiffusion, a higher Smagorinsky coefficient (compared to the equivalent QU test) is required to suppress grid-scale noise due to the mesh refinement, which, in turn, restricts the numerical stability. When the hyperdiffusion is activated, we can use a moderate Smagorinsky coefficient that does not challenge the numerical stability, and maintains a quality solution. The DTP (dycore–tracer–physics) splitting mode benefits from this most, because diffusion is called at the step of the model physics. Second, even with a higher coefficient, the Smagorinsky diffusion is inactive over certain regions where weak flow deformation dominates. In the baroclinic wave test, some grid-scale oscillations are more conspicuous over these regions (see preprint). Such noise is due to the mesh transition, and some of it is akin to the Hollingsworth instability. As will be shown, activating a background hyperdiffusion for the horizontal wind successfully removes such noise. The solutions are overall less oscillatory than those in the preprint. Third, in the tropical cyclone tests with full physics, which presents more nonlinear feedback, the hyperdiffusion option is also effective in suppressing some isolated-scale structures. Unlike the minor disturbances that may be generated near the major tropical cyclone in the simple physics test, these systems are far away from the initial vortex, and thus highly unrealistic. For these reasons, the hyperdiffusion option is used for the VR mode, but not for the QU test. Thus, QU and VR models are applied by different forms of explicit diffusion in this manuscript. Future work may also need to examine the possible impact of hyperdiffusion in a QU model to better isolate its effect.

The hyperdiffusion operator is formulated by recursively using the Laplacian operator (see Z19 for details). The diffusion coefficient can be determined in a relatively empirical way. For VR modeling, we adopt the approach documented in Zarzycki et al. (2014) for scaling the coefficient:

$$K_4(\Delta x) = K_4(\Delta x^{ref})(\frac{\Delta x}{\Delta x^{ref}})^{3.3219}; \qquad (1)$$

The reference length $\Delta x^{ref}$ and reference viscosity coefficient $K_4(\Delta x^{ref})$ are empirically determined. This formulation reduces $K_4(\Delta x)$ by a factor of 10 for every halving of resolution. A similar scaling approach is also used

---

[4] To achieve consistency with earlier QU tests.

in MPAS (Skamarock 2016). We typically use the configuration for a G-level resolution that is close to the finest resolution on the mesh. Some typical values for GRIST are documented in Table S1. These values[5] are smaller than those in Zarzycki et al. (2014) for the corresponding length scale by a factor of 5. The local grid distance ($\Delta x$) is an average distance between the grid point and all its nearest neighbors, that is, a cell-based value. The edge-based value used for hyperdiffusion is an average of two neighboring cell values.

## 3. Generation of the VR mesh

The properties and generation of the SCVT are detailed in Ringler et al. (2011) and Ju et al. (2011). We focus on two key elements of the SCVT: generators and density function. A spherical Voronoi tessellation is a spatial subdivision of a sphere $\Omega$ based on a set of distinct points on $\Omega$. For each point $x_i$, $i = 1, ..., n$, the corresponding Voronoi region $V_i$, $i = 1, ..., n$, is defined by:

$$V_i = \{x \in \Omega \mid \|x - x_i\| < \|x - x_j\| \ \ for \ j = 1, ..., n \ and \ j \neq i\}; \tag{2}$$

where $\|\cdot\|$ denotes the geodesic distance. Each point $x_i$ is called a generator and its corresponding Voronoi region $V_i$ is called the Voronoi cell. A spherical Voronoi tessellation becomes an SCVT when the generators are also the centroids of the Voronoi cells. In this study, the SCVT mesh is constructed by an iterative process based on Lloyd's algorithm (Du et al. 1999). In particular, a parallel algorithm is used (Jacobsen et al. 2013) to avoid time-consuming serial construction. That said, generating a quality VR SCVT is still nontrivial (but only done once). In our implementation, the iteration stops when two criteria are satisfied: (i) it reaches an empirically determined minimum step; and (ii) the circumcenter of each triangle falls within its shape.

### 3.1 Generators

Three original generators are used in this study:

(i) **Icosahedron bisection**. This approach benefits from the excellent uniform properties due to bisections of a regular icosahedron. The mesh resolution is referred to as G-level/G$n$, where $n$ denotes the number of bisections. After each bisection, the total grid number is approximately four times greater than the previous one.

(ii) **Icosahedron bisection with a final-step trisection**. Instead of using recursive bisection, a trisection is used at the final step, to achieve an intermediate resolution between two neighboring G-level resolutions. This mesh is referred to as G$n$B3 (i.e., $n$ bisections plus one trisection). For instance, the resolution of G5B3 (~80 km) is between that of G6 (~120 km) and that of G7 (~60 km). The number of added primal cells (mainly hexagons) from G5 to G5B3 is equal to four times the number of dual cells (triangles) in G5.

(iii) **Spherical uniform random (SUR) set of points**. Initial points are created uniformly on the sphere by the Monte Carlo method. In this way, the original generators can be obtained by using an arbitrary number of points, not restricted to a sub-divided icosahedron.

### 3.2 Density function

By specifying the density function, the SCVT is able to precisely control the distribution of the local resolution. For any two Voronoi regions indexed by $i$ and $j$, the conjecture is:

$$\frac{dx_i}{dx_j} \approx [\frac{\rho(x_j)}{\rho(x_i)}]^{1/4}; \tag{3}$$

where $\rho(x_i)$ is the density function evaluated at $x_i$, and $dx_i$ measures the local mesh resolution. This relation is valid in a theoretical sense.

A QU mesh can be constructed when the density is one on the sphere, and the Voronoi regions are approximately equivalent to each other. For the VR mesh, the basic density function used in this study is:

---

[5] Based on some tests, further halving the coefficient for a given resolution is also acceptable; see Fig. 4.

$$\rho(x_i) = \frac{1}{2(1-\gamma)}\left[tanh\left(\frac{\beta-\|x_{rc}-x_i\|}{\alpha}\right) + 1\right] + \gamma; \tag{4}$$

where $\|x_{rc} - x_i\|$ denotes the geodesic distance between the location of the refinement center and each generator. $\alpha$ indicates the width of the transition zone between the fine-resolution and coarse-resolution regions; $\beta$ defines the coverage radius of the fine-resolution region; $\gamma$ measures the densification ratio between the finest and coarsest resolutions. A sample X4 mesh based on this density function with $\gamma = (1/4)^4$ is shown in Fig. 1a.

Because the basic density function is fixed to a single-region refinement, we adjust it for multi-region refinement. The multi-region refinement is divided into two styles based on the refinement centers. In the hierarchical refinement way (Fig. 1b), we add a uniform intermediate-resolution region between the inner fine-resolution and the outer coarse-resolution regions (see Fig. 2). This helps to avoid an overly high densification ratio between two neighboring regions. Eq. (4) can be generalized to a form that allows us to control the resolution of the intermediate region:

$$\rho(x_i) = \frac{1}{2(1-\gamma)}\left[\frac{1-\lambda}{1-\gamma}tanh\left(\frac{\beta_1-\|x_{rc}-x_i\|}{\alpha_1}\right) + \frac{\lambda-\gamma}{1-\gamma}tanh\left(\frac{\beta_2-\|x_{rc}-x_i\|}{\alpha_2}\right) + 1\right] + \gamma; \tag{5}$$

$\lambda$ is designed to control the resolution of the intermediate-resolution region, also referred to as the 2nd-refinement region $(dx_{r_2})$ that is located between the 1st-refinement $(dx_{r_1})$ and the coarse-resolution regions $(dx_c)$:

$$\frac{dx_{r_1}}{dx_{r_2}} \approx \lambda^{1/4}; \tag{6}$$

Generally, $\lambda \in [\gamma, 1]$. The function of $\gamma$ is similar to that in the previous single-region refinement, except that the fine-resolution region is referred to as the 1st-refinement region here:

$$\frac{dx_{r_1}}{dx_c} \approx \gamma^{1/4}; \tag{7}$$

Corresponding to $\gamma$, we refer to the meshes that are generated based on $\lambda$ values of $(1)^4$, $(1/2)^4$, and $(1/3)^4$ as XL1, XL2, and XL3 meshes, since the resolutions of the 1st-refinement and the 2nd-refinement regions vary according to the inner densification ratios of 1, 2, and 3, respectively. For example, when $\gamma$ is fixed at X4, the hierarchical meshes based on G6 are called G6X4L1, G6X4L2, and G6X4L3 meshes.

In a polycentric refinement mode (Fig. 1c), by adding a different refinement center $x_{rc2}$, the density function of the polycentric refinement mode is defined as:

$$\rho(x_i) = \frac{1}{2(1-\gamma)}\left[tanh\left(\frac{\beta-\|x_{rc1}-x_i\|}{\alpha}\right) + tanh\left(\frac{\beta-\|x_{rc2}-x_i\|}{\alpha}\right) + 2\right] + \gamma; \tag{8}$$

The geodesic distance between the two refinement centers should satisfy $\|x_{rc1} - x_{rc2}\| > 2\beta$.

## 4. Dry-atmosphere simulations

### 4.1 Single-region refinement

The dry-atmosphere test examines the pure numerical solution of the model. It does not include the nonlinear interaction between dynamics, moisture transport, and parameterization. Based on the JW06 baroclinic wave test, we first compare the VR and QU simulations. Previous studies that employed this test for a VR model include Gettelman et al. (2018; using the multiresolution approach) and Harris and Lin (2012; using the multi-grid approach), based on different evaluation metrics. The QU grids include G6 (~120 km; 40962 cells), G7 (~60 km; 163842 cells), and G8 (~30 km; 655362 cells). The VR grids examine all three generators: (i) icosahedral bisection (e.g., G6X4), (ii) final-step trisection (G5B3X4), and (iii) spherical uniform random points (SURX4). The minimum iteration number is 300,000 for G6X4, and 1,000,000 for G5B3X4 and SURX4. The refinement center is placed at 35°N, 180°E, with $\alpha = \pi/20$ and $\beta = \pi/6$. The detailed model configuration is given in Table 1. The timestep of the VR model is limited by its fine-resolution region and is set accordingly, although the currently used timesteps do not represent the maximum allowable step.

We first examine the ability of resolving the fine-scale structures. Figure 3 shows the relative vorticity field at the model level nearest to 850 hPa (level 24) after 10 days. These values have been remapped to the Voronoi cell from the

raw triangular grid, so as to avoid the aliasing of certain oscillatory patterns. Such patterns (see preprint) actually reflect
the mesh shape, and are more conspicuous for coarse resolution. Figure S1 further shows the QU model results
interpolated to the regular longitude–latitude grid as a reference. As the resolution increases, the QU models simulate
stronger vortices with a clear filament structure (Fig. 3a–3c). The VR model can simulate the smooth structure of the
waves in the refined region, as in the high-resolution QU model. In the VR mode, two vortices in the west fall within
the fine-resolution region. The structure of the westernmost vortex in G6X4 (Fig. 3d) is close to G7. G5B3X4 and
SURX4 (Fig. 3e, 3f) further produce finer scale structures than G6X4, closer to G8. The easternmost wave falls within
the transition zone in three VR runs. The variation in the mesh sizes there does not distort the wave pattern, and the
fine-scale structure can be improved as the resolution of the transition zone increases (e.g., from G6X4 to G5B3X4).
In the SURX4 test, a minor roughness is found on the tails of the vortices. This deficiency is largely due to local mesh
irregularities. Compared to the icosahedron-based SCVT generators, the random generators tend to slightly degrade the
mesh quality and the simulation performance.
To present a more quantitative evaluation of the momentum field, we examine regional kinetic energy spectra[6]
over the refinement area (red box in Fig. 3). We use the discrete cosine transform to perform this analysis (cf. Denis
et al. 2002). The computational procedure is briefly documented in the Appendix. The decomposition is made for relative
vorticity and divergence. Fig. 4a–4c show the results from different tests. It is clear that the rotational mode dominates
the kinetic energy. The hydrostatic model (Fig. S2) produces similar results to the nonhydrostatic model; thus only the
nonhydrostatic core is discussed in the main text.
At the $1^{st}$ wavenumber, runs with different resolutions show larger discrepancies. This is because this lowest mode
absorbs most of the large-scale trend, and corresponds to only a half-cosine wave. Denis et al. (2002) suggested that
one may remove this mode if desired, but we keep it here. From the $2^{nd}$ to the $10^{th}$ wavenumber, all the curves are
consistent overall, suggesting that the well-resolved structures are robust to various tests. The major difference lies in
the high wavenumbers. For the VR results, G6X4 is close to G7. This is expected because they have similar resolution
over the selected domain (~60 km). For the same reason, G5B3X4 and SURX4 produce better spectral tails than G7
and G6X4, closer to G8. This confirms that increasing the fine resolution of the VR model is able to improve its fine-
scale resolving ability. At such high wavenumbers, SURX4 is even closer to G8 as it has slightly higher energy in the
tail. However, this actually reflects that SURX4 has slightly more grid-scale oscillations than G5B3X4. Therefore, an
examination of kinetic energy spectra should be accompanied by a close look at the real field.
In the context of kinetic energy spectra, it is useful to demonstrate the impact of the hyperviscosity coefficient.
For G5B3X4, we vary the reference hyperviscosity coefficient under a fixed reference length (30 km). Note that
$2 \times 10^{12}$ is the default configuration for this test. As shown in Fig. 4d–4f, when using $2 \times 10^{13}$, spectra are seriously
damped at the high wavenumbers in both the rotational and divergent components. A flat tail is generated. This indicates
that the diffusion strength is overly strong for the fine-resolution area. The other four tests generally produce consistent
results: reducing the coefficient tends to slightly uplift the tail, that is, increase the kinetic energy. While the lowest
coefficient ($2 \times 10^{11}$) seems to produce a nicer tail than the default run, this, again, reflects the fact that slightly more
small-scale oscillations are generated and the solutions are less clean (but still acceptable). Therefore, when tuning a
VR model, one should achieve a minimally required hyperviscosity that neither significantly damps the field nor
becomes unable to suppress certain grid-scale oscillations. For this test, $2 \times 10^{12}$ is fairly close to the optimal choice.
Next, we assess the solution errors by first examining the surface pressure field. Its global distribution on day 9
simulated by GRIST-NDC is shown in Fig. S3. The locations and magnitudes of the high- and low-pressure centers are

---

[6] Note that in Z19, due to an incorrect display setting, there is a plotting mistake in the kinetic energy spectra (their Fig. 10): the tick marks of the entire top *x*-axis (representing wavelength) do not correspond to the actual wavelength of the data. The bottom *x*-axis is correct.

consistent overall in the QU (Fig. S3a–3c) and VR (Fig. S3d–3f) runs. There are some nonzero wave patterns in all VR simulations over the Southern Hemisphere, reflecting that the nonuniform grid structure leads to higher truncation errors. G5B3X4 has smaller wave patterns than G6X4. This is expected as G5B3X4 has a higher resolution than G6X4 there. Moreover, G5B3X4 is also better than SURX4 with regard to these wave patterns. Considering that SURX4 has more mesh cells, this suggests that generators based on a regular icosahedron produce higher mesh quality than random generators, given roughly the same number of iterations.

Figure 5 shows a quantitative comparison. The global $l_2$ error norm of each test is computed against the highest resolution (G8). The errors of three VR meshes (G6X4, G5B3X4, and SURX4) are higher overall than those of the QU meshes. Note that these errors are also slightly higher than those in the preprint because a larger time step is used in this version (we performed additional tests to check this sensitivity; figure not shown). The nonhydrostatic solver produces slightly smaller errors using three VR meshes, but the overall accuracy of the two solvers is comparable. For three VR meshes, G5B3X4 shows the smallest error, and is generally close to the results of G6 during the first 10 days. G6X4 shows the largest error and SURX4 lies in between them. Again, the fact that SURX4 is less accurate than G5B3X4 implies that random generators are more likely to be trapped into the local area during the iterative procedure of mesh generation, leading to a higher degree of local irregularity.

For GRIST-NDC, we further tested G7X4 and G8X4. As shown (Fig. 5b), G7X4 produces smaller errors overall than G5B3X4, although the difference is small. G8X4 produces higher errors than G7X4 during the first four days. We suspect this is because certain initial imbalance becomes more active in a high-resolution VR configuration. Such imbalance can be caused by the discrete initialization, for example, the continuous properties imposed by the analytic wind field will not be exactly satisfied by the discrete normal velocity, unless the velocity is obtained based on some constraints. After day 5, G8X4 produces smaller errors than G7X4, and the increment in error reduction is close to the difference between G6X4 and G7X4. G8X4 also produces clearly smaller errors than G6 from day 5 to day 12. After day 13, G8X4 has higher errors than G6, although its coarsest resolution is finer than that of G6. These results suggest that the VR models indeed increase the global errors, but the errors can be reduced by increasing the coarse resolution of the mesh. This is not surprising but a reconfirmation of the conclusion in Ringler et al. (2011), using a 3D atmospheric dynamical core.

The JW06 test was originally proposed in the era that models based on a regular latitude–longitude or Gaussian grid were predominant. For a quasi-uniform grid, it is well known that the steady state in the Southern Hemisphere (or in an unperturbed condition) cannot be perfectly maintained due to mesh irregularity (Lauritzen et al. 2010). In a baroclinic environment, such errors will grow exponentially to break the steady state. Because mesh irregularity increases in a VR mode, the inability to maintain the steady state can be further amplified, ultimately contributing to the increased errors.

To reduce the impact of this issue, we add another perturbation over the Southern Hemisphere to excite two wave trains, as was done in some earlier studies (e.g., Gassmann 2013). Figure 6 shows the relative vorticity field on day 10 from three selected tests: G6, G6X4, and G5B3X4. The sign of the relative vorticity in the Southern Hemisphere is flipped to facilitate a visual comparison. The QU model (Fig. 6a) produces a comparable wave train in each hemisphere. The wave trains are not exactly symmetric because the mesh cells are not exactly symmetric across the equator. In the Northern Hemisphere, G6X4 and G5B3X4 (Fig. 6b and 6c) produce similar solutions to Fig. 3d and 3e. In the Southern Hemisphere, the solution of G5B3X4 is closer to that of G6 than G6X4, in terms of wave pattern and magnitude. This is to be expected as G5B3X4 has higher resolution in the Southern Hemisphere. Although G6X4 cannot simulate the structure in the Southern Hemisphere as G5B3X4, no serious problem is found for that region.

Figure 7 presents a quantitative estimation of the solution error. Generally, the magnitude and the growth of the errors are close to those in Fig. 5b. A notable difference is that G8X4 produces smaller relative errors, closer to those

of G7 from day 8 to day 12. On day 15, G8X4 still shows comparable errors to G6. The error reduction from G6X4 to
G7X4 also becomes larger than that in Fig. 5b, when the waves become more developed (e.g., day 8 to 11). This implies
that the inability to maintain a steady state in the Southern Hemisphere indeed worsens the error estimation for a VR
model.

**4.2 Multi-region refinement**

To increase the application range of the single-region refinement, two multi-region refinement modes are examined
to obtain desired resolutions in multiple regions. This represents a unique aspect of the multiresolution approach. The
first way is a hierarchical style with one refinement center. It contains three consecutive uniform sub-regions outside
the refinement center: the $1^{st}$-refinement region, the $2^{nd}$-refinement region, and the coarse-resolution region. The $2^{nd}$-
refinement region provides an intermediate resolution between the $1^{st}$-refinement and coarse-resolution regions.
Compared to the single-region refinement, this $2^{nd}$-refinement region provides more uniform resolution between the
refinement center and the coarse-resolution area.
The symmetrical perturbation test was performed using a G8X4L2 mesh. On day 10, the first wave (the strongest)
over the Northern Hemisphere is experiencing a higher gradient of resolution than that over the Southern Hemisphere.
The fine-scale structures are well captured by the VR model (Fig. 8a). The second and third waves in the Northern
Hemisphere have stronger magnitudes than their equivalents in the Southern Hemisphere. The difference in each wave
train generally reflects the differences in the local resolution.
The second way uses a polycentric style with multiple refinement centers. The same double-baroclinic wave trains
are generated using a G7X4 mesh, with two refinement centers at 35°N, 180°E and 35°S, 180°E, respectively. On day
10, the model well simulates the fine-scale structures over two selected regions (Fig. 8b). The transitions between the
fine and coarse regions are smooth and stable. This refinement mode may provide an effective way to simultaneously
improve the simulations over different regions. The wave train in the Southern Hemisphere coincides well with that in
the North. Again, they are not exactly identical because the mesh cells are not exactly symmetric across the equator.
In this test case, it is useful to demonstrate the impact of activating the hyperdiffusion option. As shown in Fig. 8c
and 8d, when hyperdiffusion is turned off, the simulation results become rather oscillatory. The noise pattern at the tail
of the wave is akin to the Hollingsworth instability, and is amplified by a VR model due to mesh transition. The noise
is more conspicuous in the $1^{st}$-refinement region of G8X4L2, due to higher resolution and rapid mesh transition. Also
note that these results are more oscillatory than those in the preprint, because the Smagorinsky diffusion in that version
is stronger. As mentioned in Section 2.3, even so, the Smagorinsky option is still inactive over certain regions. Using a
background hyperdiffusion is a good complement to this deficiency.

**5. Moist-atmosphere modeling**

**5.1 Simple physics**

Moist-atmosphere modeling includes the nonlinear interaction of dynamics, moisture transport, and model physics.
We use the idealized tropical cyclone test (Reed and Jablonowski 2011) because of its clear resolution sensitivity. This
test is useful to examine the VR performance because the solution does not fully converge even at 10 km (Z20). The
simulated tropical cyclone is very sensitive to the mesh size. A higher resolution model will produce more intense
storms. An initial vortex is placed 10°N, 180°E with no background flow. The tropical cyclone moves northwestward
due to beta drift. The initial virtual temperature profiles are designed to be conditionally unstable in the troposphere. A
small perturbation (either physical or computational) is more likely to excite new storms. Whether these additional
signals are realistic depend on the situations.
Previous studies have shown that model dynamics has a clear impact on the simulations of tropical cyclones. For
example, Zhao et al. (2012) showed that increasing the strength of 2D divergence damping in the finite-volume
dynamical core (Lin 2004) leads to more occurrences of tropical cyclones. Reed et al. (2015) showed that, in CAM5,

the spectral element core produces stronger tropical cyclones than the finite-volume core, when the parameterization suite remains almost unchanged. For GRIST, a known sensitivity is that different tracer transport options may lead to different wind magnitudes in the eyewall, as the full pressure gradient term is, by design, related to tracer mixing ratios (Z20).

We first examine the model behaviors under a simple-physics environment (Reed and Jablonowski 2012). In the preprint, we compared two representative groups of numerical tests: one group based on the NDC with DTP splitting enabled; and one based on the HDC with no DTP splitting (i.e., dycore, tracer transport, and physics use the same time step). Results from these two groups are consistent overall. The impacts of DTP splitting and hydrostatic/nonhydrostatic options do not generate discernable differences across various VR meshes. This is consistent with the previous QU model tests (Z19; Z20) in that: (i) the nonhydrostatic solver behaves similarly to its hydrostatic counterpart under the hydrostatic regime; and (ii) the DTP splitting does not cause a degeneration in performance when it is properly configured. In this version, which uses an updated configuration, only the NDC is tested. The configuration for the simple physics test is given in Table 2.

We first examine the evolution of the tropical cyclone when it moves across the transition zone in the hierarchical refinement mode. The mesh is fixed at X4, with $\alpha_2 = \pi/36$ and $\beta_2 = \pi/4$. In the control run (G6X4L2; Fig. 9a), $\alpha_1 = \pi/36$, $\lambda = (1/2)^4$, and $\beta_1 = \pi/12$. The 1st-refinement region of this VR mesh has ~40-km resolution. Two cases are used to examine the impact of the mesh distribution. In the first case, we choose to use more rapid resolution changes in the inner transition zone based on the two parameters $\alpha_1$ and $\lambda$. $\alpha_1$ controls the width of the inner transition zone, and $\lambda$ represents the densification ratio between the 1st- and 2nd-refinement regions. Either narrowing the width of the transition zone (Fig. 9b) or enlarging the inner densification ratio (Fig. 9c) generates a more abrupt transition zone. The other way is to have the transition zone affect the tropical cyclone at an earlier stage. The initial cyclone is placed closer to or even partly within the transition zone by increasing $\beta_1$ (Fig. 9d), a parameter that controls the radius of the 1st-refinement region. The tropical cyclone is initialized at 10°N, 180°E over the 2nd-refinement region, near the transition zone between the 1st-refinement and the 2nd-refinement regions.

On day 10, all four tests well simulate the shape of the tropical cyclone. During its movement from the 2nd-refinement into the 1st-refinement region, the change in the grid size leads to little distortion on the tropical cyclone. Compared to the preprint, two major differences can be found. First, the minor disturbances near the major tropical cyclones almost disappear in this version, because activating hyperdiffuion damps the wind field more effectively. Though it can be damped, this minor disturbance is not that unrealistic because it is near the major cyclone. During the movement, it is possible that the nonlinear feedback can generate new small-scale systems. Similar minor disturbances have also been observed in some models participating in DCMIP2016 (see e.g., https://www.earthsystemcog.org/projects/dcmip-2016/). The other difference is that the maximum wind magnitude in the eyewall is stronger overall in this version. This is due to the locally scaled Smagorinsky formulation, as mentioned in Section 2.3.1.

Figure 10 presents the evolution of the tropical cyclone in each test on days 2, 4, 6, and 8. The tropical cyclones in all tests are consistent. No discernable difference is found when they move across the transition. A slightly larger difference is more evident in the early stage (day 2), but such differences diminish as the tropical cyclones move into the refinement center. The relative vorticity field also looks smooth and does not show any artifacts (Fig. S4). This indicates that the mesh transition does not create notable problems.

To further examine possible sensitivity, we performed a group of experiments by altering one of the three parameters: $\alpha_1$, $\beta_1$, and $\lambda$. Figure 11 shows the minimum surface pressure (Fig. 11a–11c) and the maximum wind speed at 850 hPa (Fig. 11e–11f). All these tests use a DTP splitting number 1 : 4 : 8, with a dycore step of 60 s (Table 2). Only one test with the smallest $\alpha_1$ ($\frac{\pi}{90}$) is unstable at this DTP splitting number, so we adjusted it to 1 : 2 : 4 with the same dycore

step. This suggests that an overly narrow inner transition zone can impose a higher stability restriction. The tropical cyclone rapidly strengthens during the first two days. After day 2, it enters into a moderately developing stage in each experiment. The evolution of intensity is diverse. All the VR runs tend to produce stronger tropical cyclones than G6 or G7, because the fine-resolution area determines the ultimate strength. The tests have run-to-run differences, but they are small overall, showing consistent results and robustness.

Figure 12 shows a further comparison between two VR runs (G6X4L2 and G7X4L2) and two QU runs (G7 and G8). The highest resolution of G6X4L2 and G7X4L2 is slightly higher than that of G7 and G8, respectively (see Table 2). As shown, in each comparison (G6X4L2 vs G7, G7X4L2 vs G8), the VR model produces stronger wind magnitudes in the eyewall and a more compact size featuring a smaller area coverage. The eyewall of the cyclone converges towards its center, almost within 1 degree from the center. The maximum winds develop to higher vertical levels as the local resolution increases. Overall, the difference between the VR and QU tests is attributed to different local resolutions.

In the tropical cyclone test, if an unscaled Smagorinsky eddy viscosity is used (Fig. S5; preprint), the VR model will show a higher parametric sensitivity for the Smagorinsky coefficient than the QU model. The current version has largely reduced such sensitivity, closer to the behavior of the QU model (Fig. S5). This confirms the reasonable behavior of the local length formulation.

**5.2 Full physics**

The simple physics test, while insightful, is limited in the sense that the physics processes are simplified and do not support enough of the nonlinear feedback typical of a real-world model. A VR model may introduce a higher degree of nonlinear feedback due to mesh refinement. Thus, it is useful to check its behavior given a full parameterization suite. We use a more highly variable mesh: G6B3X16L4. The finest part on this mesh reaches ~7–10 km. The refinement center is placed at 35°N, 165°E. The sea surface temperature is identical to that in the simple physics test (29℃ uniformly). The starting date is the first day of June. The solar constant is 1370 W.m$^{-2}$. The DTP splitting number is 1 : 5 : 10 with a dycore step of 10 s. If activated, the square of the Smagorinsky coefficient is 0.005, and the reference hyperviscosity is $2 \times 10^{10}$ m$^4$.s$^{-1}$ with a reference length scale of 7000 m.

We first focus on the impact of the explicit diffusion process. Figure 13 shows the tropical cyclones on day 10 from three tests: no explicit diffusion (noDiff), using hyperdiffusion only for the horizontal velocity (hyper), and hyperdiffusion plus Smagorinsky (hyper+smg). The wind speed is shown at the model level nearest to 850 hPa (level 24). On day 10, the tropical cyclone center just moves into the refinement center (35°N, 165°E). In general, the results in the three tests are consistent. The major difference lies in the eyewall. The noDiff and hyper tests produce slightly higher wind maxima than hyper+smg. This suggests that Smagorinsky diffusion becomes active in the eyewall, diffusing the solutions a little bit more strongly. Except for this minor difference (see the zoomed results), hyper and hyper+smg produce very close solutions.

Explicit diffusion, while it seems to be irrelevant to the performance over the fine-resolution region, plays a more important role in the coarse-resolution region. Due to the globally uniform sea surface temperature, near the South Pole, some isolated systems can be generated in the noDiff test (Fig. 14a). Unlike the possible minor disturbances near the major tropical cyclone in the simple physics test, these systems are not expected there because they are far away from the initial vortex. They are likely to be caused by the nonlinear feedback between dynamics and physics. Only using the Smagorinsky diffusion can suppress them to a certain extent, but not enough due to its flow-dependent nature (Fig. 14b). By further activating the hyperdiffusion option, these systems can be effectively removed (Fig. 14c). This suggests that the explicit diffusion configuration is able to achieve a balance between underdiffusion and overdiffusion.

With this X16 mesh, it is also useful to examine whether a single refinement with a high densification ratio (G6B3X16) tends to create serious problems during the simulation, especially when the tropical cyclone moves across the transition. The refinement center is rotated to 60°N, 165°E. The tropical cyclone will encounter more mesh

transitions in G6B3X16. The results are shown in Fig. 15. On day 2, the tropical cyclone in G6B3X16L4 is experiencing mesh cells between ~30 km and ~50 km. In G6B3X16, the encountered mesh sizes are above 70 km. Hence, G6B3X16L4 has a stronger and more developed eyewall at this stage. On day 6, the eyewall enters into a region with cell sizes ~25–30 km in G6B3X16L4, and ~50 km in G6B3X16. G6B3X16L4 still has a more compact eyewall, but the difference between the two runs becomes smaller. On day 10, the eyewalls in both tests enter into an area with cell sizes ~20–25 km. At this stage, both cyclone systems show similar maximum wind magnitudes in the eyewall, and exhibit a similar distribution. Clearly, the fine-scale structure is determined mainly by the fine resolution, although two systems can have larger differences before they enter into the fine-resolution area. These results also demonstrate that a single refinement with a high densification ratio, though more challenging, can perform competitively with a more gradually refined mesh when properly configured (e.g., an overly narrow transition is undesirable). The choice of different mesh styles thus demands more numerical modeling experience and depends on the target issue. This needs to be further examined with more realistic weather and climate system.

**6. Conclusions**

In this study, an atmospheric model formulated on an unstructured mesh has been systematically configured and evaluated in its VR mode. Different mesh-refinement styles have been utilized to evaluate the model performance under increasing degrees of complexity, from dry dynamics to simple and then to full physics. Based on some additional NDC tests with full physics, it is found that the QU and VR models (with the same degrees of freedom) have comparable parallel efficiency, that is, the VR configuration does not cause a degeneration in strong scaling performance. This is to be expected as the domain decomposition does not distinguish between QU and VR grids. Other scaling performance (e.g., whether the speedup ratio can achieve the ideal value by simply reducing the total grid number, with all other things being equal) can be found in Liu et al. (2020). Overall, these results demonstrate that the VR configuration of GRIST is a reliable and economic alternative to high-resolution quasi-uniform modeling. The adverse impact due to the mesh transition and the coarse-resolution area can be well controlled. The major conclusions may be summarized in three aspects.

**On the overall performance.** Based on the dry baroclinic wave test, all VR styles can well capture the fine-scale wave structures. Such fine-scale resolving capability is supported by analysis of regional kinetic energy spectra, and is further verified in the multi-region refinement mode. In the transition zone, the waves are not adversely affected by the mesh refinement. In the coarse-resolution region, the VR model can also simulate an equivalent distribution of waves to its low-resolution counterpart. A VR model indeed produces greater solution errors compared to its QU counterpart that has the same degrees of freedom. However, the solution error can be reduced when the coarse region increases its resolution. Thus, its impact can be controlled. In the tropical cyclone test, the VR model in simple and full physics can simulate the gradual evolution of the tropical cyclone, showing reasonable resolution sensitivity. Importantly, the simulation of the fine-scale structure is controlled mainly by the fine resolution, although larger differences may exist before the systems move into the refinement area.

**On the impact of explicit diffusion.** Comparing this version with its earlier version (preprint), the impact of explicit diffusion can be clearly demonstrated. It has been shown that activating scaled hyperdiffuion for the horizontal velocity as background diffusion is helpful in alleviating grid-scale oscillations due to mesh transitions. It also suppresses some highly unrealistic disturbances found in the full physics scenario. When activating hyperdiffuion, it is suggested that a minimally required hyperviscosity should be used, such that it neither significantly damps the field nor becomes unable to suppress undesirable grid-scale disturbances. Using only Smagorinsky diffusion would require higher coefficients to remove oscillations, which, in turn, would restrict the numerical stability. Even so, the Smagorinsky-only configuration is not enough for the VR mode, because it becomes inactive over certain regions that are dominated by weak flow deformation. It is also shown that, in the tropical cyclone test, the scaled Smagorinsky diffusion has much lower

parametric sensitivity than its unscaled counterpart. In general, the use of explicit diffusion (properly scaled) plays a
positive role in the VR configuration, although there were concerns that their own discretization may introduce
additional problems. The increased demands of explicit diffusion from the QU to VR configuration is in accordance
with increasing mesh discontinuity due to transitions, and thus should not be viewed as a disadvantage.
**On the impact of the mesh styles.** Different mesh styles are all able to produce fine-scale structures. They maintain
solution quality over the mesh transition and the coarse-resolution area. In the dry test, it has been shown that G5B3X4,
while it has fewer cells than SURX4, leads to smaller solution errors and less oscillatory solutions. As these two meshes
are generated given a similar number of iterations, this *might* suggest that the random generators need more computation
to achieve a quality mesh. In general, when generating a VR mesh, we empirically suggest that one would be better to
start from a subdivided icosahedron. In the tropical cyclone case, a series of sensitivity tests examined the model
performance in a hierarchical refinement style. The solutions exhibit consistency even when the VR mesh is slightly
perturbed by one of the three parameters that control the density function. It is suggested that an overly narrow transition
zone should be avoided. In the full physics test, G6B3X16L4 and G6B3X16 produce consistent results on day 10,
although the simulations at the initial stage (e.g., day 2) exhibit larger differences.
**Code and data availability**: GRIST is available at https://github.com/grist-dev. A version of the model code, and
running and postprocessing scripts for supporting this paper are available at: https://zenodo.org/record/3930643.
Version 2 is for this manuscript, and version 1 is for the preprint. The preprint is available from the online link of this
paper. The grid data used to enable the tests are located at https://zenodo.org/record/3817060. Public code access is
available after authorization, following https://github.com/GRIST-Dev/TermsAndConditions.
## Appendix: Regional kinetic energy spectra analysis
The model data are first interpolated to the Gaussian grid. For the QU model, we use the Gaussian grid that is
close to the nominal resolution of the icosahedral mesh (T106 for G6, T213 for G7, and T426 for G8). For the VR
model, the data are interpolated based on the fine resolution of the mesh (T221 for G6X4, T328 for G5B3X4, and T328
for SURX4). The selected regional domain (red box in Fig. 3) ranges from 150°E to 150°W and from 30°N to 65°N,
with two vortices.
We follow the approach documented by Denis et al. (2002) for computing kinetic energy spectra on a selected
regional domain. Let $F_\zeta(m,n)$ be the discrete cosine transform (DCT) of a 2D relative vorticity field $\zeta(i,j)$ of $N_i$-
by-$N_j$ grid points. The variance array can be computed from the DCT field as:

$$\sigma_\zeta^2(m,n) = \frac{F_\zeta^2(m,n)}{N_i N_j}, \tag{A1}$$

where $m = 0, 1, 2, ..., N_i - 1$, $n = 0, 1, 2, ..., N_j - 1$, and $(m,n) \neq (0,0)$. Each 2D wavenumber pair $(m,n)$ is
associated with a wavelength:

$$A = \frac{2\Delta}{\mu}, \tag{A2}$$

where $\Delta$ is the grid spacing. $\mu$ is a normalized 2D wavenumber defined as:

$$\mu = \sqrt{\frac{m^2}{N_i^2} + \frac{n^2}{N_j^2}}. \tag{A3}$$

To construct a spectrum, the variance contributions of $F_\zeta(m,n)$ needs to be binned by bands of $\mu$. The ranges of $\mu$
for a given wavelength band are determined as follows:

$$\mu(k) = \frac{k}{\min(N_i, N_j)}, \tag{A4}$$

$$\mu(k) + \Delta\mu(k) = \frac{k+1}{\min(N_i, N_j)}, \tag{A5}$$

where $k = 1, 2, 3, ..., \min(N_i, N_j) - 1$ denotes the wavenumber.
The rotational part of a spectrum as a function of wavenumber $k$ can be obtained by binning each element in the

variance array. For a given $k$, Eqs. (A4) and (A5) are used to calculate the lower and upper bounds. For each variance element, if the corresponding $\mu$ satisfies $\mu(k) < \mu < \mu(k) + \Delta\mu(k)$, the variance will be summed to give the spectrum for this $k$, that is:

$$E_\zeta(k) = \frac{1}{2}\sum_{\mu(k)}^{\mu(k)+\Delta\mu(k)} \frac{\sigma_\zeta^2(m,n)}{\hat{k}^2}, \tag{A6}$$

where $\hat{k}$ is the circular wavenumber:

$$\hat{k} = \frac{\pi}{\Delta}\mu. \tag{A7}$$

Similarly, the divergent part can be evaluated based on the divergence field:

$$E_D(k) = \frac{1}{2}\sum_{\mu(k)}^{\mu(k)+\Delta\mu(k)} \frac{\sigma_D^2(m,n)}{\hat{k}^2}, \tag{A8}$$

where $\sigma_D^2(m,n)$ is the variance array of divergence. The total kinetic energy as a function of wavenumber $k$ is given as:

$$E(k) = E_\zeta(k) + E_D(k), \tag{A9}$$

In this study, kinetic energy spectra are displayed as a function of wavelength $A$ and wavenumber $k$. It should be noted that the lowest mode ($k = 1$) absorbs most of the large-scale trend and can be excluded if needed (cf. Denis et al. 2002).

**Supplement**: vr_supplement.pdf contains Table S1 and Figures S1–S5.

**Author contribution:** Y. Zhou performed the mesh generation, most data analysis, and an initial exploration of VR modeling, with input and guidance from Y. Zhang. R. Yu supervised Y. Zhou's work, provided impetus and resources, guided experimental design. J. Li supervised the project of model development. Z. Liu is responsible for parallel computing, and customized the parallel mesh generation software. Y. Zhang designed this study, developed and maintained the model, performed production runs, and analyzed the simulations. Y. Zhang and Y. Zhou wrote this manuscript, with contributions from all authors. All the authors continuously discussed the model development.

**Competing interests.** The authors declare that they have no conflict of interest.

**Acknowledgments.** The constructive comments from two anonymous reviewers are highly appreciated. This study was supported by the National Natural Science Foundation of China (41875135; 41675075), the National Key R&D Program of China (2017YFC1502202; 2016YFA0602101), and the S&T Development Fund of CAMS (2019KJ011).

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

**Table 1: Mesh resolution, timesteps, and diffusion coefficients for the dry-atmosphere test.**

| Test case | Mesh | Mean/min/max averaged cell distance (km) | HDC: timestep (s) | NDC: timestep (s) | Square of Smagorinsky coefficient ($c_s^2$) | $K_4(\Delta x_{ref})$ $m^4 s^{-1}$ | $\Delta x_{ref}$ $m$ |
|---|---|---|---|---|---|---|---|
| Baroclinic wave: single perturbation | G6 QU | 120.17/97.93/121.76 | 150 | 150 | - | - | - |
| | G7 QU | 60.09/47.60/60.88 | 90 | 90 | 0.015 | - | - |
| | G8 QU | 30.04/23.04/30.54 | 60 | 60 | 0.04 | - | - |
| | G6X4 | 107.93/42.95/188.08 | 40 | 40 | 0.015 | $2 \times 10^{12}$ | 30000 |
| | G5B3X4 | 71.98/27.55/125.74 | 40 | 40 | 0.015 | $2 \times 10^{12}$ | 30000 |
| | SURX4 | 68.27/28.34/119.65 | 40 | 40 | 0.015 | $2 \times 10^{12}$ | 30000 |
| | G7X4 | 53.96/19.88/93.71 | Not tested | 40 | 0.015 | $1 \times 10^{12}$ | 20000 |
| | G8X4 | 27.04/9.84/46.85 | Not tested | 20 | 0.015 | $2 \times 10^{11}$ | 15000 |
| Baroclinic wave: double perturbations | G6 QU | 120.17/97.93/121.76 | Not tested | 150 | - | - | - |
| | G7 QU | 60.09/47.60/60.88 | 90 | 90 | 0.015 | - | - |
| | G8 QU | 30.04/23.04/30.54 | 60 | 60 | 0.04 | - | - |
| | G6X4 | 107.93/42.95/188.08 | Not tested | 40 | 0.015 | $2 \times 10^{12}$ | 30000 |
| | G5B3X4 | 71.98/27.55/125.74 | Not tested | 40 | 0.015 | $2 \times 10^{12}$ | 30000 |
| | SURX4 | 68.27/28.34/119.65 | Not tested | 40 | 0.015 | $2 \times 10^{12}$ | 30000 |
| | G7X4 | 53.96/19.88/93.71 | Not tested | 40 | 0.015 | $1 \times 10^{12}$ | 20000 |
| | G8X4 | 27.04/9.84/46.85 | Not tested | 20 | 0.015 | $2 \times 10^{11}$ | 15000 |
| | G8X4L2 | 28.80/13.14/61.19 | Not tested | 20 | 0.015 | $2 \times 10^{11}$ | 15000 |
| | G7X4-polycentric | 53.02/26.63/114.94 | Not tested | 40 | 0.015 | $2 \times 10^{12}$ | 30000 |

**Table 2: Mesh resolution, timesteps, and diffusion coefficients for the simple-physics test.**

| Test case | Mesh | | | Mean/min/max averaged cell distance (km) | NDC: timestep (s) | Square of Smagorinsky coefficient ($c_s^2$) | $K_4(\Delta x_{ref})$ $m^4 s^{-1}$ | $\Delta x_{ref}$ $m$ |
|---|---|---|---|---|---|---|---|---|
| Tropical cyclone: base test | G6 QU | | | 120.16/97.08/121.83 | 150-600-1200 | - | - | - |
| | G7 QU | | | 60.08/47.09/60.92 | 90-360-720 | 0.015 | - | - |
| | G8 QU | | | 30.04/23.04/30.54 | 60-240-480 | 0.015 | - | - |
| | G6X4L2 | | | 113.24/40.23/166.00 | 60-240-480 | 0.01 | $8 \times 10^{12}$ | 40000 |
| | G7X4L2 | | | 56.62/19.71/82.64 | 30-120-240 | 0.01 | $8 \times 10^{11}$ | 20000 |
| Tropical cyclone: sensitivity to the three parameters of the hierarchical refinement mode (various hierarchical refinement meshes based on the G6X4 mesh) | $\alpha_1$ | XL $(\lambda)$ | $\beta_1$ | | | | | |
| | $\frac{\pi}{90}$ | 2 | $\frac{\pi}{12}$ | 113.81/39.40/163.52 | 60-120-240 | 0.01 | $8 \times 10^{12}$ | 40000 |
| | $\frac{\pi}{60}$ | 2 | $\frac{\pi}{12}$ | 113.62/39.35/165.12 | 60-240-480 | 0.01 | $8 \times 10^{12}$ | 40000 |
| | $\frac{\pi}{45}$ | 2 | $\frac{\pi}{12}$ | 113.40/39.61/165.68 | 60-240-480 | 0.01 | $8 \times 10^{12}$ | 40000 |
| | $\frac{\pi}{36}$ | 2 | $\frac{\pi}{12}$ | 113.24/40.23/166.00 | 60-240-480 | 0.01 | $8 \times 10^{12}$ | 40000 |
| | $\frac{\pi}{30}$ | 2 | $\frac{\pi}{12}$ | 113.02/40.71/165.23 | 60-240-480 | 0.01 | $8 \times 10^{12}$ | 40000 |
| | $\frac{\pi}{36}$ | 1.5 | $\frac{\pi}{12}$ | 110.59/44.58/179.62 | 60-240-480 | 0.01 | $8 \times 10^{12}$ | 40000 |
| | $\frac{\pi}{36}$ | 2.5 | $\frac{\pi}{12}$ | 114.26/37.10/158.05 | 60-240-480 | 0.01 | $8 \times 10^{12}$ | 40000 |
| | $\frac{\pi}{36}$ | 3 | $\frac{\pi}{12}$ | 114.72/36.77/153.26 | 60-240-480 | 0.01 | $8 \times 10^{12}$ | 40000 |
| | $\frac{\pi}{36}$ | 3.5 | $\frac{\pi}{12}$ | 114.73/36.74/150.61 | 60-240-480 | 0.01 | $8 \times 10^{12}$ | 40000 |
| | $\frac{\pi}{36}$ | 2 | $\frac{\pi}{9}$ | 111.42/39.30/170.05 | 60-240-480 | 0.01 | $8 \times 10^{12}$ | 40000 |
| | $\frac{\pi}{36}$ | 2 | $\frac{5\pi}{36}$ | 109.65/40.54/177.41 | 60-240-480 | 0.01 | $8 \times 10^{12}$ | 40000 |
| | $\frac{\pi}{36}$ | 2 | $\frac{\pi}{6}$ | 107.82/43.03/186.09 | 60-240-480 | 0.01 | $8 \times 10^{12}$ | 40000 |
| Tropical cyclone: sensitivity to the Smagorinsky coefficient ($C_s$) | G6 QU | | | 120.16/97.08/121.83 | 150-600-1200 | 0.005/0.0075/0.01/0.015/0.02 | - | - |
| | G6X4L2 | | | 113.24/40.23/166.00 | 60-240-480 | 0.005/0.0075/0.01/0.015/0.02 | $8 \times 10^{12}$ | 40000 |

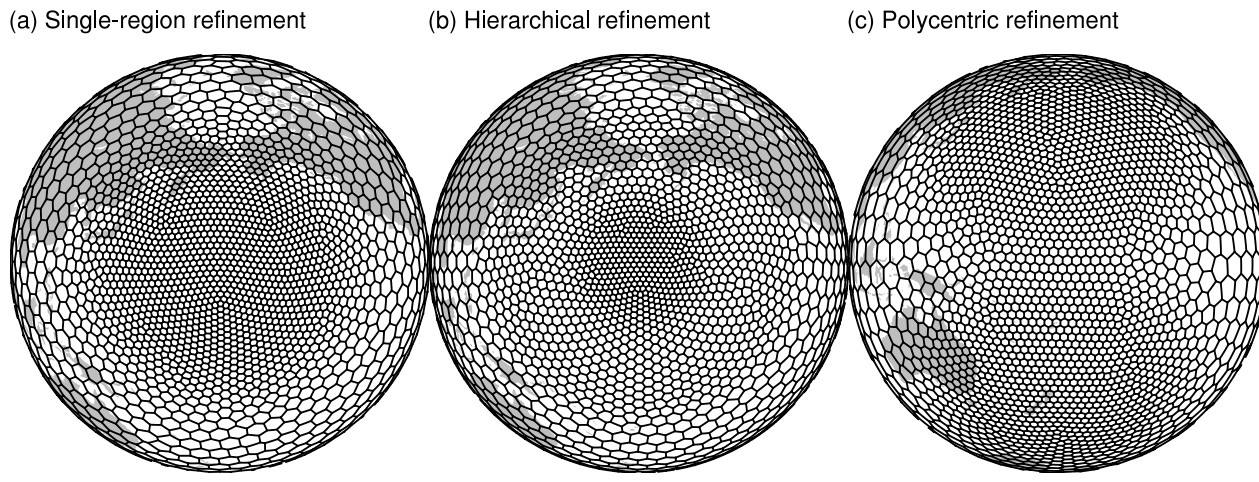

(a) Single-region refinement   (b) Hierarchical refinement   (c) Polycentric refinement


Figure 1: An illustration of the variable-resolution mesh (X4) based on three density functions: (a) single-region
refinement, (b) hierarchical refinement, and (c) polycentric refinement. Throughout this paper, all land-sea outlines
are only given for a spatial reference, and do not represent the geographical difference.

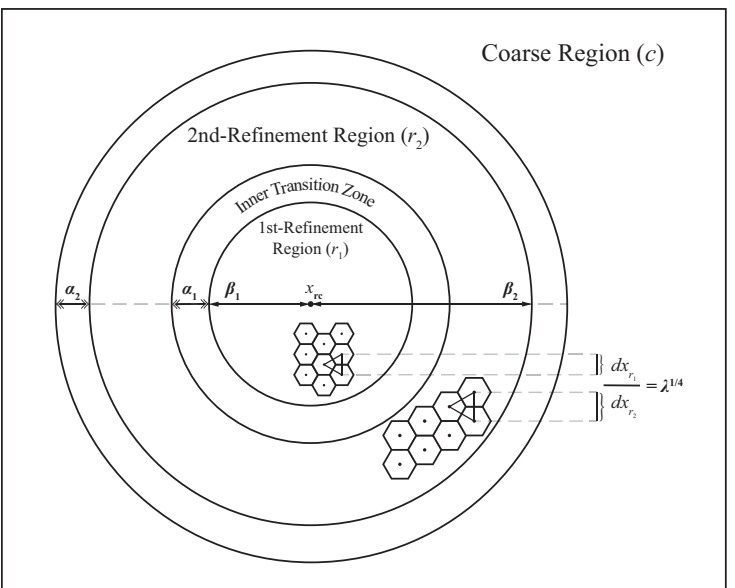


Figure 2: A schematic diagram of the hierarchical refinement mesh, illustrating the function of three parameters of the
density function. $\alpha_1$ and $\alpha_2$ control the width of the transition zones; $\lambda$ represents the inner densification ratio
between the 1$^{st}$-refinement and 2$^{nd}$-refinement resolution regions; $\beta_1$ and $\beta_2$ control the coverage radius of the 1$^{st}$-
refinement and 2$^{nd}$-refinement regions.

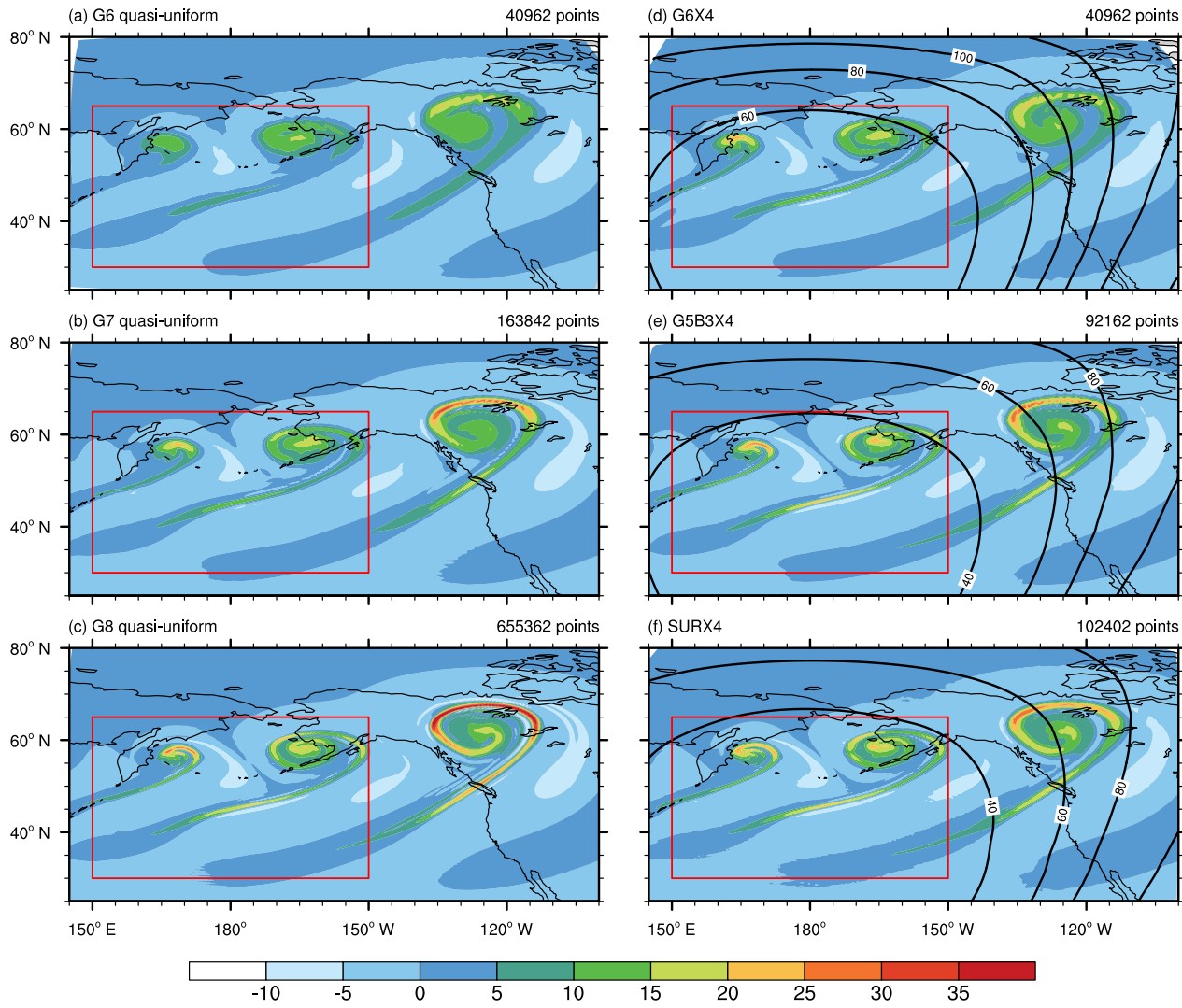

Figure 3: The relative vorticity ($10^{-5}$ s$^{-1}$) field at the model level nearest to 850 hPa (level 24) after 10 days. The quasi-uniform (left-hand column) and variable-resolution (right-hand column) results are shown. The contour lines denote the mesh resolution (km). The vorticity is remapped from the raw triangular cell to the Voronoi cell using an area-weighted approach (true for all vorticity values shown in this study). Also see Figure S2 for the remapped QU-model solutions on a regular latitude–longitude grid. The red box denotes the region for regional kinetic energy analysis in Fig. 4. The contour lines denote the smoothed mesh cell sizes (km).

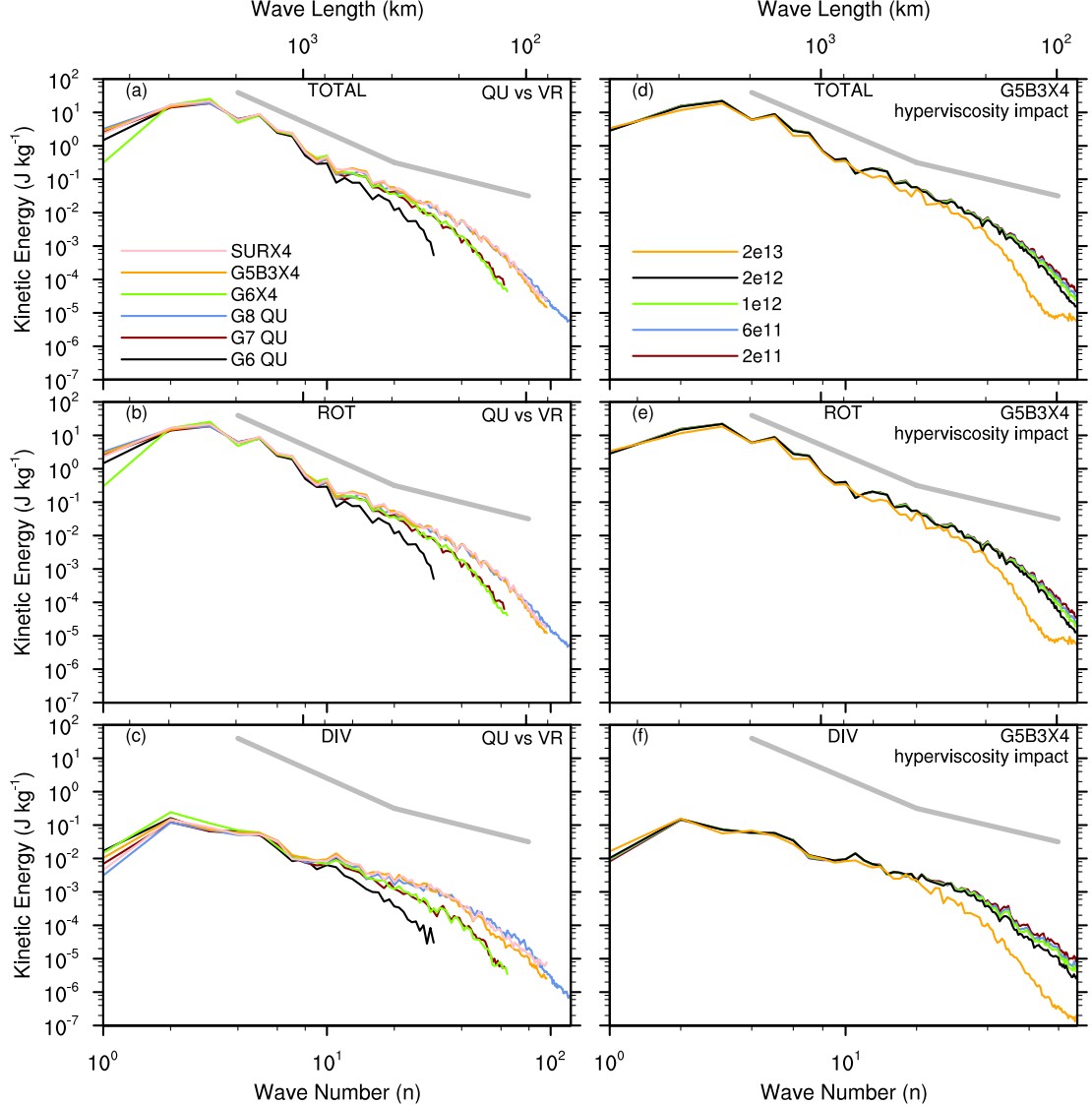


Figure 4: Baroclinic wave: horizontal regional kinetic energy spectra (unit: J kg⁻¹) at the model level nearest to 850
hPa on day 10, (a) total kinetic energy, (b) the rotational component, and (c) the divergence component. The results of
the nonhydrostatic core are shown here, while the results for the hydrostatic core can be found in Fig. S2. The right-
hand column (d)–(f) examines the impact of varying the hyperviscosity coefficient on spectra. The thick gray lines
denote the −3 and −5/3 slopes, respectively.

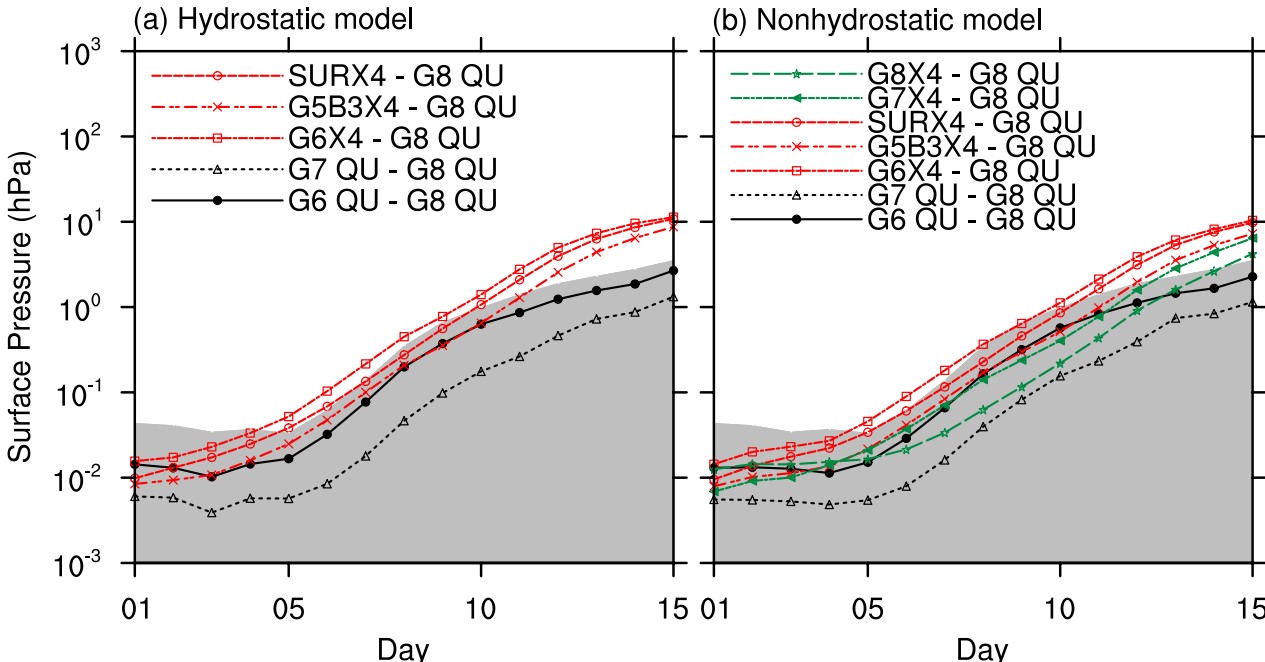

Figure 5: Baroclinic wave test: the $l_2$ error norms of surface pressure as a function of time for (a) the hydrostatic and (b) the nonhydrostatic dynamical core. The error is computed against the high-resolution quasi-uniform G8 mesh. The gray area denotes the uncertainty in the reference solutions, which selects the maximum $l_2$ error norms from four curves (HDC and NDC at G7 and G8 as four cases). Details can be found in Jablonowski and Williamson (2006).

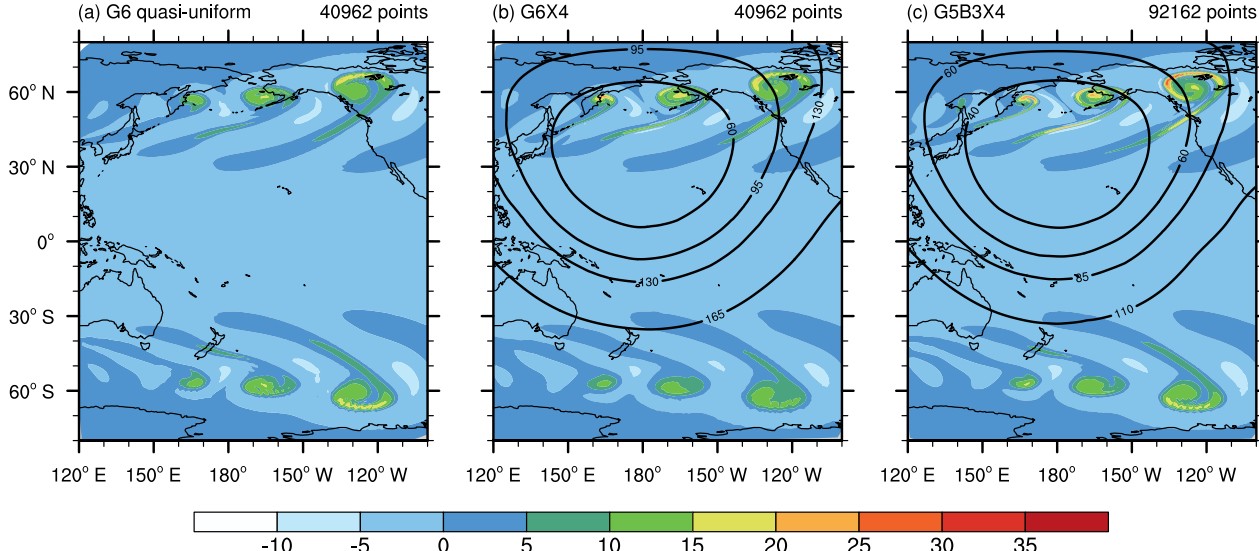

Figure 6: Adding a symmetrical perturbation in the Southern Hemisphere in the baroclinic wave test. The relative vorticity ($10^{-5}$ s$^{-1}$) field at the model level nearest to 850 hPa on day 10 is shown for (a) quasi-uniform G6, (b) variable-resolution G6X4, and (c) G5B3X4 meshes. Only the nonhydrostatic core is used. The values in the Southern Hemisphere are substituted by their opposite values. The raw vorticity values on the triangular cell have been remapped to the Voronoi cell. The contour lines denote the smoothed mesh cell sizes (km).

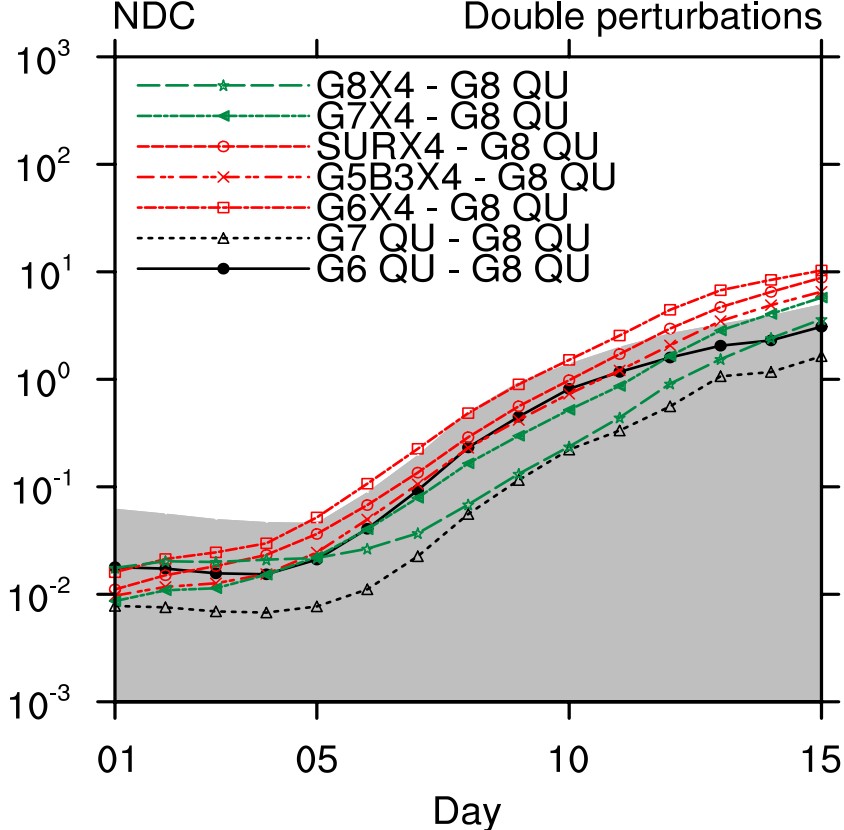

802

803    Figure 7 Same as Fig. 5, but for the baroclinic wave test with double perturbations, only for the nonhydrostatic core.
804

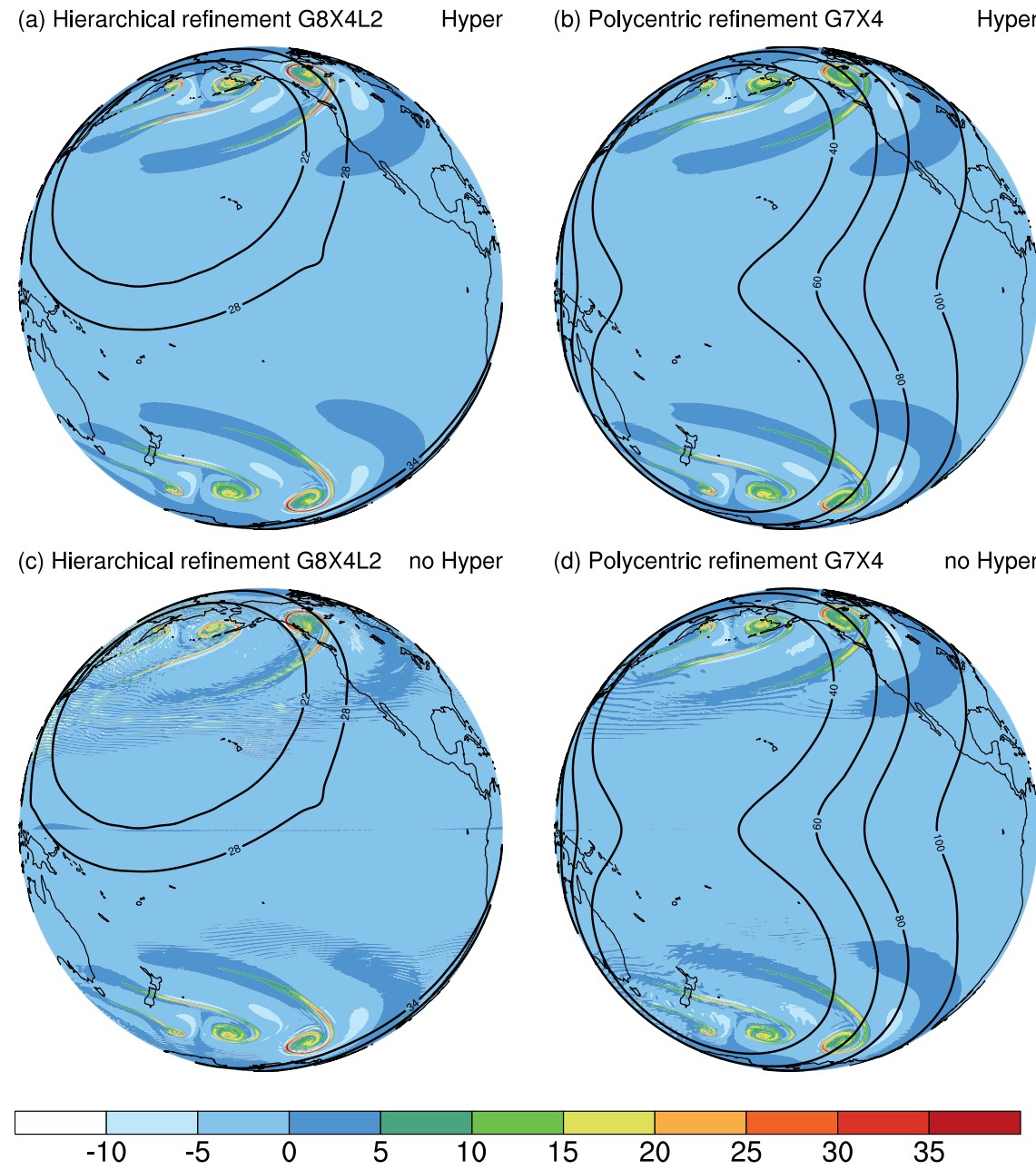

Figure 8: As in Fig. 6, but for two multi-region refinement meshes: (a) a hierarchical refinement style (G8X4L2) and (b) a polycentric refinement mesh based on G7X4. (c) (d) same as (a) (b), but for runs that turn off the hyperdiffusion option. The contour lines denote the smoothed mesh cell sizes (km). Note that the small kink in the 28 km contour of (a) (c) simply reflects that the generated mesh occasionally has higher local irregularities at some areas.

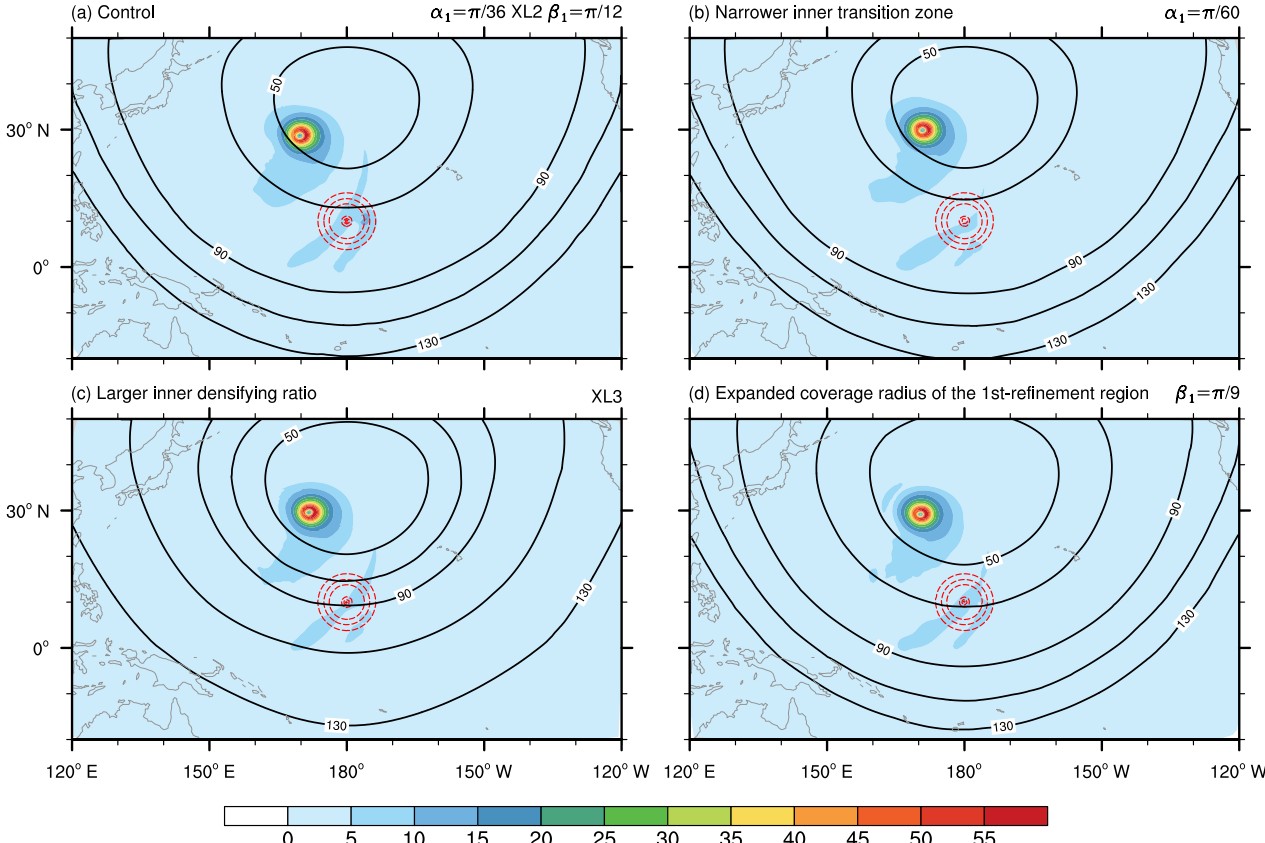

Figure 9: Idealized tropical cyclone test: the horizontal wind speed (m s$^{-1}$) at 850 hPa after 10 simulation days based on hierarchical refinement meshes with (a) the control, (b) reduced $\alpha_1$, and (c) higher $\lambda$ for more rapid changes in the transition zone, and (d) larger $\beta_1$ to make the transition zone affect the tropical cyclone in an earlier stage. The red dashes denote the initial location of the tropical cyclone. The contour lines denote the smoothed mesh cell sizes (km).

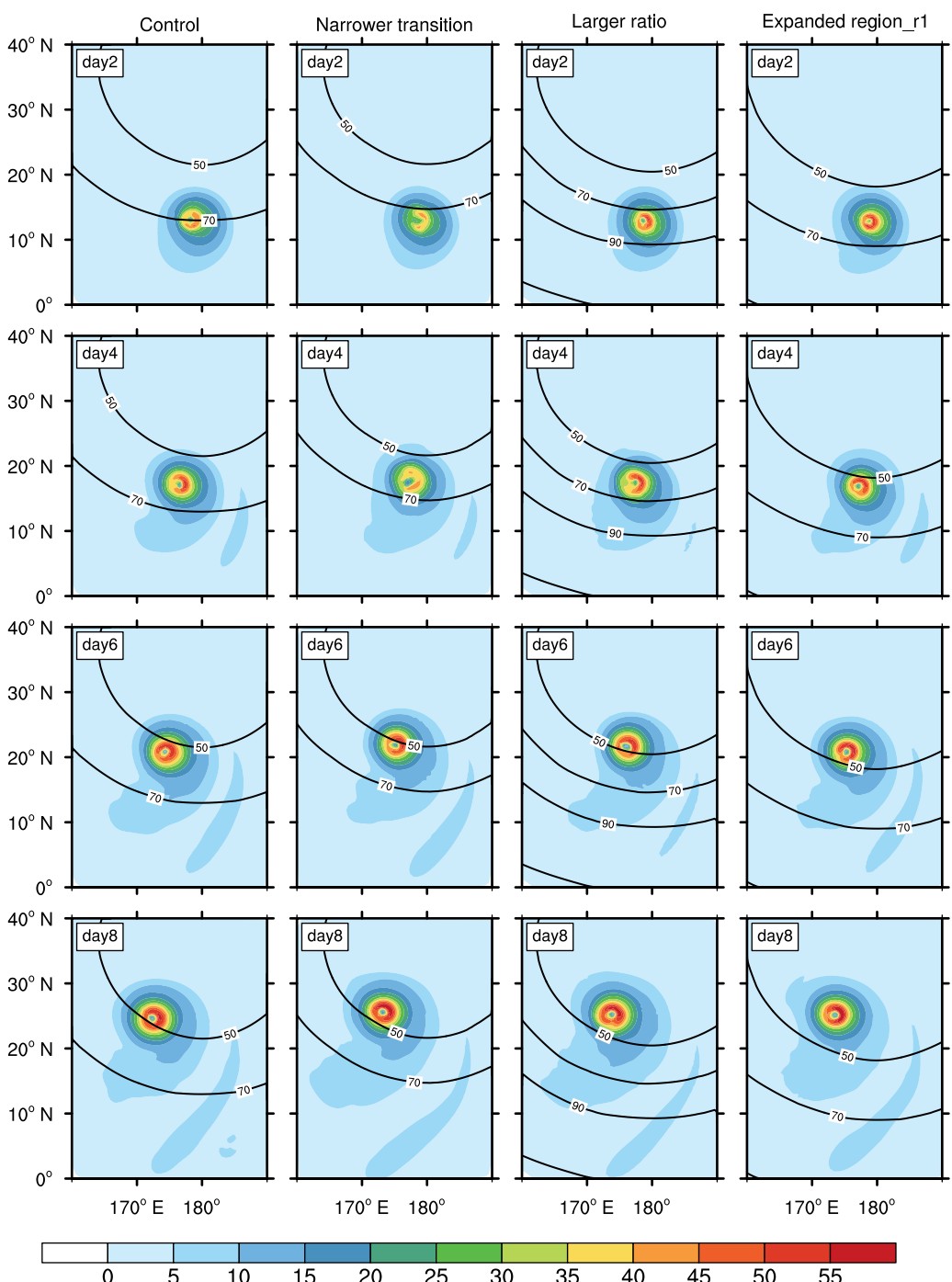


Figure 10: Corresponding to the four cases shown in Fig. 9, the simulation results on days 2, 4, 6, and 8 are shown to
examine the movement of the tropical cyclones when they cross the mesh transition.

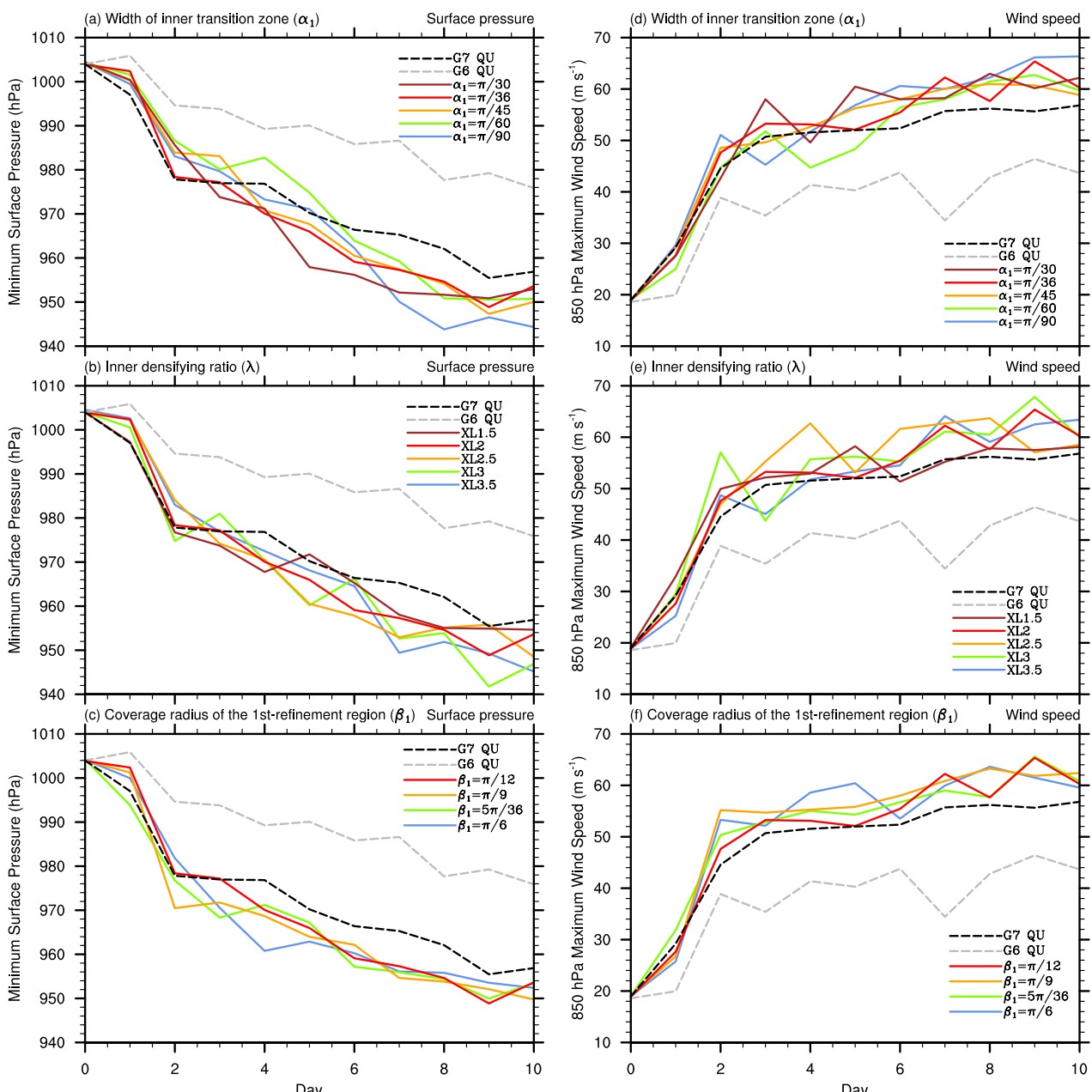


Figure 11 Idealized tropical cyclone test: temporal evolution of minimum surface pressure and maximum horizontal
wind speed at 850 hPa based on the quasi-uniform G6 and G7 meshes, and the hierarchical refinement meshes by
varying (a, d) $\alpha_1$, (b, e) $\lambda$, and (c, f) $\beta_1$. When one parameter is altered, the other two parameters are fixed as in the
control run ($\alpha_1 = \pi/36$, $\lambda = (1/2)^4$, and $\beta_1 = \pi/12$). The three mesh parameters control the width of the inner
transition zone, the inner densification ratio, and the coverage radius of the 1st-refinement region.

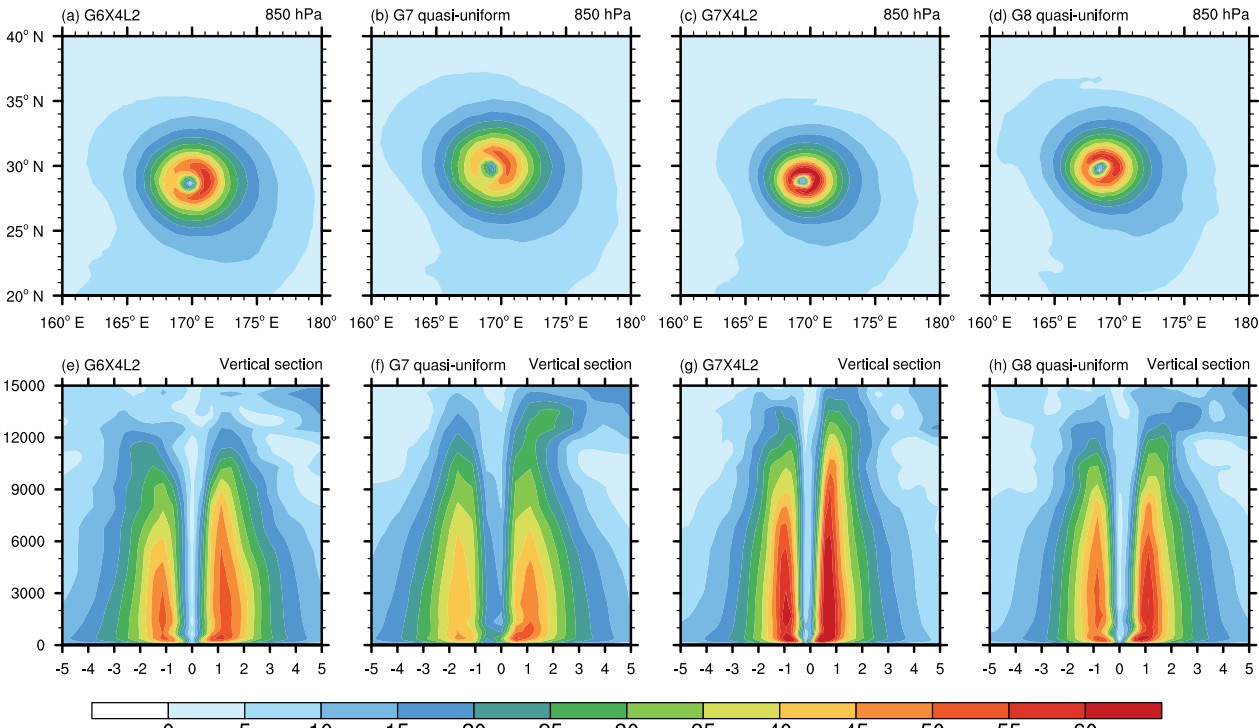

Figure 12: Idealized tropical cyclone test: (a–d) the horizontal wind speed (m s$^{-1}$) at 850 hPa after 10 simulation days based on quasi-uniform and variable-resolution meshes and (e–h) the corresponding vertical cross-section of the wind speed with a meridional range of ±5-degrees from the center of the tropical cyclone. The vertical coordinate of the cross-section denotes the height (m).

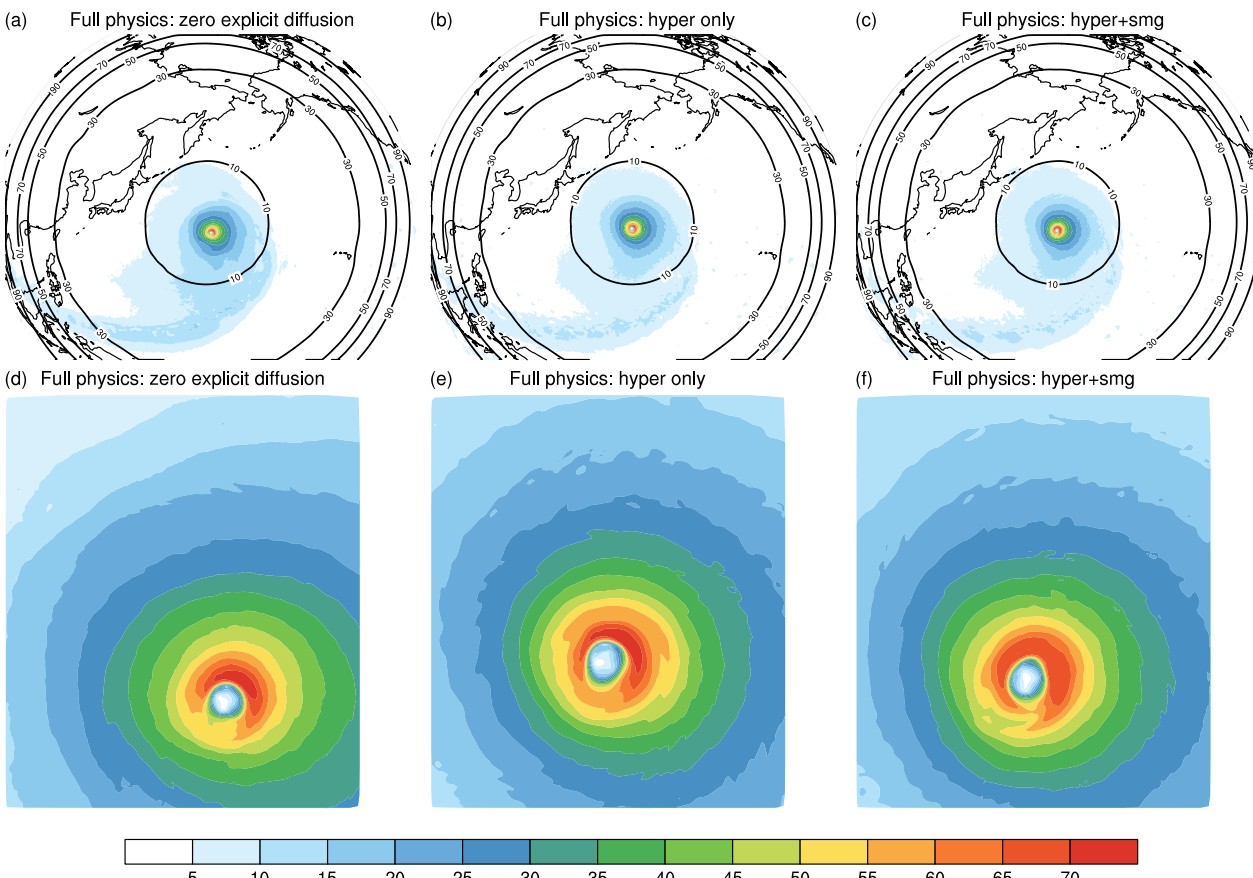

Figure 13 GRIST-NDC full physics tests with the G6B3X16L4 mesh: (a) no explicit diffusion is used, (b) only hyperdiffusion for the horizontal velocity is used, (c) both hyperdiffusion and Smagorinsky diffusion are used. (d)-(f) same as (a)-(c), but zooming in on the major cyclone system. The results are shown for the wind speed (m.s$^{-1}$) at the model level nearest to 850 hPa (level 24) on day 10. The contour lines denote the smoothed mesh cell sizes (km).

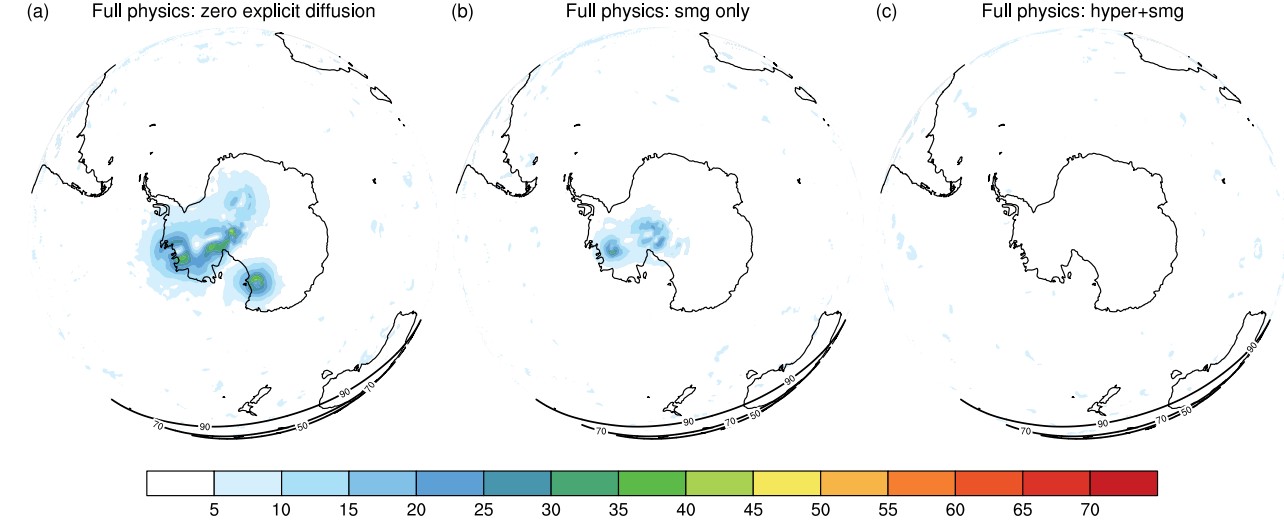


Figure 14: Same as Fig. 13, but the map is rotated to the South Pole.

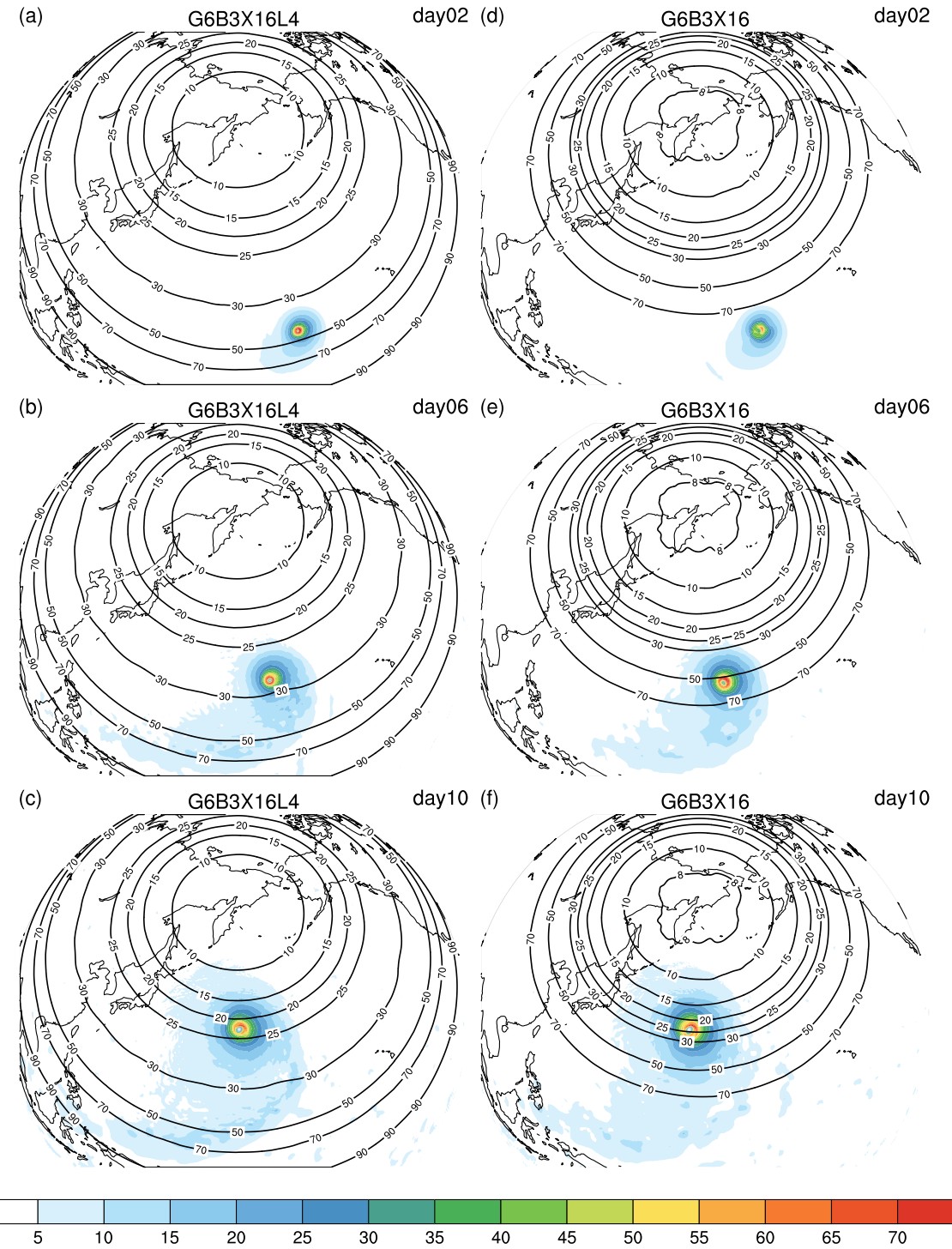


Figure 15: The same tests as in Fig. 13, but the refinement center is rotated to 60°N, 165°E. The left-hand column
shows the results on days 2, 6, and 10 on the G6B3X16L4 mesh, the right-hand column shows the corresponding
results on the G6B3X16 mesh. The results are shown for the wind speed (m.s⁻¹) at the model level nearest to 850 hPa
(level 24).