# Peer review of "Configuration and Evaluation of a Global Unstructured Mesh Atmospheric 1"

_Geoscientific Model Development, 2020_

## Referee Comment (RC1) · Anonymous Referee #1 · 6 Aug 2020

**Review of 'Configuration and Evaluation of a Global Unstructured Mesh Model based on the Variable-Resolution Approach'**

In this manuscript, the variable-resolution (V-R) version of the GRIST model, based on Voronoi tessellations is described. The authors define the mesh generation process and explore multiple refinement approaches with a dry dynamical core test case (the Jablonowski-Williamson baroclinic wave) and a moist test case with simplified physics (the Reed-Jablonowski tropical cyclone). They subjectively (visually) and objectively ($l_2$ errors, etc.) compare V-R simulations against quasi-uniform (Q-U) reference simulations and verify the V-R simulations perform generally as one would expect, particularly based on previous findings with V-R models using similar test cases.

V-R models have indeed been shown to be useful tools and multiple modeling centers are currently pursuing their development. Therefore, further evaluation and validation of such configurations is warranted, especially as V-R models become more commonly used for scientific research and application.

In general, the results here are a confirmation of robust performance rather than any overtly new physical insight. This makes GMD a suitable venue for such work. I do find the manuscript fairly underdeveloped, however. The model description is lacking, particular describing options specific to V-R dynamical cores such as scale-specific diffusion and model timestep. The simulations evaluating the ability of the tropical cyclone to move between resolutions are interesting but feel almost tacked on, with weak expansive discussion, particularly with regard to diffusion behavior. Some other useful and commonly-reported information is also omitted, such as computational scaling numbers.

While the manuscript wasn't illegible by any means, it did contain numerous grammatical errors that detract at times from the science.

I suggest **major revisions**. Again, this is more of an application of existing test cases to an existing model to essentially demonstrate that a V-R configuration is not performing poorly. For this to be a useful reference to other users of GRIST in the future, as well as a comparison benchmark for other modeling centers, some additional evaluation and breadth of discussion is warranted.

**Major comments**

- The model description in Section 2 is lacking.

    - For example, it is unclear exactly what numerics are being applied. Finite volume, I assume? What is the vertical discretization? How close are the numerics to the Model for Prediction Across Scales (MPAS)?

    - It seems reasonable that the timestep of a global V-R simulation scales with the finest grid spacing to satisfy the CFL constraint, although this isn't explicitly stated. It would be helpful to note this, however, as some ill-posed V-R configurations can actually be more restrictive from a stability perspective than their equivalent Q-U counterparts.

    - What is the vertical resolution of the model? Is this constant across all configurations, or correspondingly increased in either/both the V-R and higher-resolution Q-U runs? How does this compare to other models with published baroclinic wave and tropical cyclone test results?

    - Appealing aspects of V-R modeling are the computational savings when solving a regional problem. Do the authors have scaling numbers that could provide a more objective quantification of this? Should they expect the simulations to scale linearly with the number of degrees of freedom in the mesh? Is there additional overhead associated with refinement that causes this scaling to be sub-linear?

- Along this line, it is unclear what (if any) modifications are made for the V-R configurations relative to the Q-U. A Smagorinsky diffusion is applied in the horizontal. Is there any additional scale-selective

explicit diffusion such as hyperdiffusion, or does the flow-dependent Smagorinsky handle everything? The latter would imply a fairly diffusive scheme in an implicit sense.

- The moist tropical cyclone test section is underdeveloped.

  – A couple sentences of additional description are warranted. What is the surface configuration, what does the idealized moist physics consist of? Convection? Boundary layer parameterization? Surface fluxes? How else is the model initialized?
  – The cyclone moves through the mesh – how is this done? Is there a background flow or does the configuration relay on beta drift associated with gradients of Coriolis across the cyclone?
  – I would postulate that relative vorticity would be a better quantity to evaluate when assessing potential distortion or wave reflection in a numerical accuracy sense (e.g., Figs. 7-8). Are there artifacts in this field during the TC transit?
  – Other relevant citations which could help contextualize the TC results with respect to dynamical core and diffusion are Zhao et al. [2012] and Reed et al. [2015].

- It is quite unclear exactly what the authors are showing in Fig. 12. Is the goal of this figure to show that V-R simulations are more sensitive to diffusion coefficient than a Q-U grid with the same setting(s)? In some ways, it is a natural finding that a cyclone transiting multiple grid spacing will 'feel' multiple diffusion scales, although as noted above, it isn't stated whether this diffusion explicitly scales with resolution or this is an implicit response. Further, in the abstract, the authors note that this 'suggest[s] the importance of parameter tuning,' although there is not enough description of the configuration to support this statement. Is this tuning just one 'number' for the whole mesh? I would recommend spending another paragraph or two explaining the importance of this finding in the context of the V-R validation exercise and how it pertains to the evaluated version of GRIST.

- Is is unclear from the KE spectra how well the V-R runs are doing. For example, they could be accumulating spurious energy near the grid cell. It doesn't appear that they are from the spatial plots, however, the interpolation to the T106 Gaussian grid means nothing definitive can be said about the refined regions within the nests since those are below the truncation scale. There are a few ways to evaluate KE spectra within a regional model or regional patch, such as those proposed by Errico [1985] and Skamarock [2004]. I would recommend their exploration.

**Minor comments**

- Lines 49-53. Wave reflection can be strongly influenced by other parameters than transition zone width, such as numerical method and grid staggering. See Ullrich and Jablonowski [2011].

- It is unclear why both hydrostatic and non-hydrostatic cores are exercised here. Both test cases do not emphasize non-hydrostatic dynamics (being of relatively 'coarse' resolution compared to regional weather models), so it should be expected that both solutions look similar in the absence of some sort of erroneous formulation. There is nothing inherently wrong with testing both cores, although it is mentioned more frequently than probably necessary.

- Section 2.2.2. is quite long, specific, and doesn't add a ton of 'added value' to the manuscript. I would recommend shortening this slightly; keeping the description of important parameters (e.g., $\gamma$, $\lambda$, etc.) and removing extraneous text.

- Lines 192-193. Why is the iteration number different for these grid methods? Is there a quantitative reason, or was this a subjective design choice during mesh generation?

- I am not sure what this sentence means in the code and data availability section: 'GRIST is available at https://github.com/grist-dev, in private repositories. A way is provided for the editor and reviewers to access the code, which does not compromise their anonymity (to our best effort).' I would double-check that this all conforms to GMD's policies.

- I believe both the baroclinic wave and tropical cyclone test case were part of the Dynamical Core Model Intercomparison Project (DCMIP) test suite. It may be worth reviewing multi-center reviews (such as Ullrich et al. [2017]) or references from other labs to see if there is any benefit in comparing results to those previously published using the same test cases.

- Fig. 2. It is not 100% clear which (X4) mesh is being shown. Are all the V-R meshes so similar they functionally look like this? If they are not, the three different generator meshes should be plotted.

- Figs. 5, 8, and 9. Why do the black refinement isolines look 'jagged?' I assume the plotting software is struggling with cell areas right at a given threshold, would recommend smoothing for visualization.

- Fig. 6. Recommend moving the reference slopes above the spectra so that they do not intersect the raw data.

**Typographical errors and grammar**

As noted above, there are numerous – albeit generally minor – grammatical errors. This list is not meant to be exhaustive, but rather, a few obvious catches I noted while reading. I recommend a thorough proofread for grammar before resubmission.

- Line 37. ... while permitting...

- Line 43. ... while retaining or minimally degrading...

- Line 64. ... maintains tropical cyclones...

- Line 83. 'three difference initial point sets' is awkward phrasing.

- Line 95. ... developmental ... (?)

- Line 216. ... model level nearest to...

- Lines 232-233. 'Nevertheless...' sentence is awkward.

- Lines 252. Perhaps something like 'sign of the relative vorticity is flipped to account for hemispheric differences' or thereabouts.

**References**

R. M. Errico. Spectra computed from a limited area grid. *Monthly Weather Review*, 113(9):1554–1562, 1985.

K. A. Reed, J. T. Bacmeister, N. A. Rosenbloom, M. F. Wehner, S. C. Bates, P. H. Lauritzen, J. E. Truesdale, and C. Hannay. Impact of the dynamical core on the direct simulation of tropical cyclones in a high-resolution global model. *Geophysical Research Letters*, 42(9):3603–3608, 2015. doi: 10.1002/2015GL063974.

W. C. Skamarock. Evaluating mesoscale NWP models using kinetic energy spectra. *Monthly Weather Review*, 132:3019–3032, 2004.

P. A. Ullrich and C. Jablonowski. An analysis of 1d finite-volume methods for geophysical problems on refined grids. *Journal of Computational Physics*, 230(3):706–725, 2011.

P. A. Ullrich, C. Jablonowski, J. Kent, P. H. Lauritzen, R. Nair, K. A. Reed, C. M. Zarzycki, D. M. Hall, D. Dazlich, R. Heikes, C. Konor, D. Randall, T. Dubos, Y. Meurdesoif, X. Chen, L. Harris, C. Kühnlein, V. Lee, A. Qaddouri, C. Girard, M. Giorgetta, D. Reinert, J. Klemp, S.-H. Park, W. Skamarock, H. Miura, T. Ohno, R. Yoshida, R. Walko, A. Reinecke, and K. Viner. DCMIP2016: A review of non-hydrostatic dynamical core design and intercomparison of participating models. *Geoscientific Model Development Discussions*, pages 1–49, 2017. doi: 10.5194/gmd-2017-108.

M. Zhao, I. M. Held, and S.-J. Lin. Some counterintuitive dependencies of tropical cyclone frequency on parameters in a GCM. *Journal of the Atmospheric Sciences*, 69(7):2272–2283, 2012. doi: 10.1175/JAS-D-11-0238.1.

---

## Short Comment (SC1) · 12 Aug 2020

GMDD-2020-150 Short Reply to Reviewer#1

Yi Zhang

The authors are very grateful to this Reviewer for their valuable comments. We will carefully improve the manuscript, add more details and experiments, and rewrite some parts. This will take some time. But as GMDD has this unique interactive discussion, I would like to first address some issues raised by this Reviewer for some clarification.

**1. The model description in Section 2 is lacking.**

**– For example, it is unclear exactly what numerics are being applied. Finite volume, I assume? What is the vertical discretization? How close are the numerics to the Model for Prediction Across Scales (MPAS)?**

Reply: We apologize for all the incompleteness. We will add more details regarding the numerical operators. In short, GRIST is formulated on an unstructured Voronoi-Delaunay mesh based on the staggering finite-volume method. This choice is made to achieve a balance of solution accuracy, efficiency, implementation and runtime cost. As a new global model group that focuses on weather-climate modeling, GRIST used some well-established techniques available in the icosahedral-/Voronoi-mesh modeling community, based on publicly available papers and documents. These details can be clearly found in the previous model description paper (Zhang et al. 2019; Zhang et al. 2020), and we will concisely summarize them in the revision. MPAS pioneered some key numerical features, and GRIST used some of them. However, some detailed formulations are clearly different. GRIST has its own unique aspects as the numerical operators are implemented under a different solution strategy and a different general environment (i.e., governing equations, vertical coordinates, physics-dynamics coupling workflow, and infrastructure). The comparison of numerics and its behaviors may be more meaningful in some isolated and highly idealized tests (e.g., passive 2D/3D advection and shallow water waves) that specifically examine the numerical operators.

**2. On several comments about the diffusion option.**

Reply: We will add more details regarding the diffusion operators. In the initial submission, the VR configuration only alters the mesh file, the timestep, and some tuning of the Smagorinsky coefficient (one for the whole mesh). There is no additional horizontal or vertical filters, except those implicitly generated by the numerics (e.g., the upwind flux operator). In the code (ParGRIST-A20-0705), there is a 4th-order computational hyperdiffusion for the horizontal wind field, but not activated for those tests. Also note that the cyclone tests activate Smagorinsky for tracer transport, which is actually *fairly unnecessary* (shape-preserving filter is enough for tracer, while the impact of activating this is rather small). This option was preserved for the supercell tests with constant-coefficient 2nd-order diffusion, and can be switched to a Smagorinsky-style diffusion in other tests. Due to the evolutionary nature of model development, the

running script does not turn it off because of regression and sanity check.

The Smagorinsky diffusion, though stronger than hyperdiffusion, does not really generate *that diffusive* solutions because of its flow dependent nature (i.e., its diffusion strength is acceptable). This nature makes it selective in terms of where and how much to damp (see e.g, Fig. 9 in Gassmann 2013, QJRMS). In contrast, the artificial computational diffusion or 2D divergence damping is always active. The weakly diffusive evidence can be clearly observed in the JW baroclinic wave solution at G8 resolution (with Smagorinsky activated), which produces very sharp gradient and filament structures for the vorticity field. The Smagorinsky diffusion is indeed stronger if fully activated, and the side effect probably lies in a slightly higher stability restriction.

The original Smagorinsky formulation works well for the tests in the initial submission, as the mesh transition is at most X4. In some recent VR modeling tests with full-physics, the original formulation (using a global mean constant length scale) is found to be unstable for the more highly-deformed mesh (~6 km-~30km-~120 km). We are testing some modifications to the original Smagorinsky: reducing the Smagorinsky coefficient, and/or using a variable length scale. Meanwhile, only using 4[th]-order hyperdiffusion (requires some code changes from ParGRIST-A20-0705) for horizontal winds or using ZERO explicit diffusion can produce reasonable solutions. Results from these three configurations look similar (Fig. 1). In full physics modeling, the physics is stronger than simple physics, so we are also going to check whether using ZERO explicit diffusion will work well for the full-physics situations. In the pure dynamical core (baroclinic wave) or simple physics (tropical cyclone) tests, using explicit Smagorinsky diffusion will in general, make the solutions look better. Also note that explicit diffusion is often used as a cleaner for dynamics (with or without physical meanings), but the discrete numerical operators may introduce additional problems, especially for the highly deformed meshes.

[Figure]

Fig. 1 GRIST-NDC with full physics in the same tropical cyclone test, using a mesh ranges from ~6 km-~30 km-~120 km (approximately estimated); (a) only-Smagorinsky for winds and potential temperature with a variable length scale and a small coefficient $c_s^2 = 0.0025$; (b) only 4th-order hyperdiffusion for the horizontal wind field with a constant coefficient; (c) no explicit diffusion. Day 10 results are shown as in the initial manuscript.

3. **Appealing aspects of V-R modeling are the computational savings when solving a regional problem. Do the authors have scaling numbers that could provide a more objective quantification of this? Should they expect the simulations to scale linearly with the number of degrees of freedom in the mesh? Is there additional overhead associated with refinement that causes this scaling to be sub-linear?**

Reply: We will check this issue. Dr. Z. Liu actually has a separate manuscript specifically focusing on the computational performance, including the VR mesh. So in this work, we will restrict ourselves to the physical performance (but we will also mention this point). For QU and VR grids with the same degree of freedom, their respective parallel efficiency will be similar, as the domain decomposition uses the same philosophy that does not distinguish between VR and QU meshes.

Moreover, we would like to point out that a VR model is definitely more cheaper than its fine-resolution QU counterpart. This advantage has clear implications, and is especially valuable for model

development. Via VR, we may economically test and examine full-physics configured GRIST at convection-permitting (CP) resolution in a *global* environment, to check whether the configuration is suitable. A global ~5 km QU icosahedron-based mesh has 23592962 primal cells. It is apparently crazy and inefficient to test and run global CP modeling at such high-resolution in our daily model development and debugging efforts. The computational resource is a key limitation. With the VR approach, one may achieve regional ~5 km with a grid number like 368642. Given the same theoretical time step and vertical levels, this implies a ~64X savings for one model variable. Moreover, the VR approach provides a more challenging environment in terms of scale variation, and the multiscale behavior of model physics can be well examined. A properly formulated VR model thus gives a valuable guidance for the fine-resolution QU model.

In short, the added value of a VR model not only lies in its application end. For model developers, it is an economical tool for developing and evaluating scale-aware physics, and an important intermediate step before establishing global CP modeling.

**4. I am not sure what this sentence means in the code and data availability section: 'GRIST is available at https://github.com/grist-dev, in private repositories. A way is provided for the editor and reviewers to access the code, which does not compromise their anonymity (to our best effort).' I would double-check that this all conforms to GMD's policies.**

Reply: GRIST is open to the general public, while needs authorization. This is a requirement in the current model development projects. Both the GitHub repo and the Zenodo link currently require authorization for access. In the initial submission, a GitHub account is provided for public access, but this way is not recommended any more. An accessible Zenodo shared link is generated, and I have asked the handling Editor to send it to all the reviewers if possible. The Zenodo link does not compromise the anonymity of access, which is required by GMD's policy. In the revision, the latest version will be uploaded as a reference. Some statements in the code and data section will be modified accordingly.

---

## Short Comment (SC2) · 14 Aug 2020

Dear authors,

in my role as Executive editor of GMD, I would like to bring to your attention our Editorial version 1.2:

https://www.geosci-model-dev.net/12/2215/2019/

This highlights some requirements of papers published in GMD, which is also available on the GMD website in the 'Manuscript Types' section:

http://www.geoscientific-model-development.net/submission/manuscript_types.html

[Figure]

In particular, please note that for your paper, the following requirement has not been met in the Discussions paper:

- "The main paper must give the model name and version number (or other unique identifier) in the title."

Please add both model name or acronym and its version number to the title of your manuscript upon revision.

Yours,

Astrid Kerkweg

---

## Referee Comment (RC2) · Anonymous Referee #2 · 4 Sep 2020

Review of *Configuration and Evaluation of a Global Unstructured Mesh Model based on the Variable-Resolution Approach*

Zhou et al.

**General Impressions**

This study evaluates the performance of the variable-resolution configuration of a newer global model GRIST, and seeks to understand the various strengths and weaknesses of different refinement meshes. The authors provide results from both dry and moist idealized experiments that illustrate that the solution in the refined regions resemble the uniform high-resolution solutions. While this take home message is clear, I would like to see further analysis/discussion on why the errors tend to be larger in VR compared with the uniform resolution runs, examples that I point out specifically in the comments section, and also how the Smagorinsky operators are implemented in VR. After addressing these minor revisions, I think this manuscript is acceptable for publication in GMD.

**Comments**

L64: CAM has multiple dycores, each with distinct numerical properties, and so this statement can be misleading. I think the authors should consider mentioning that the Zarzycki study cited used the spectral-element dycore.

L88: This statmentt "[a] series of numerical tests was performed to examine the model reliability under more challenging conditions," reads like there are more chellenging tests than the TC test-case, but the TC test-case is the most complex case used in this study.

L108: If I recall correctly, the Smagorinsky coefficients scale with grid spacing. Is the density function used to determine the Smagorinsky coefficients?

Model and configurations: Can the authors include the number of vertical levels used in the simulations?

L160: The authors argue that the densification ratio should be no larger than 1:4, and point to a citation that I can't seem to get access to. I'm having trouble interpreting this statement. Do the authors mean *no less* than 1:4? Would this then mean the refined grid spacing should be no less than a 1/4 of the coarser region grid spacing? If so, I can think of many spectral-element VR studies that use a much smaller ratio without having reported any serious errors. I could be misunderstanding entirely here, but I think this densification ratio and implications of some lower limit should be spelled out more clearly for the general reader.

L197: The authors keep referring to grid imprinting in this paragraph. Am I to infer that they are only talking about the spurious waves being generated in the southern

hemisphere, in the coarse region of the grid? These features seem to become less noisy when the coarse region increases its resolution, as one would expect. I think it should be stated that the coarser region of G5B3X4 is higher resolution than the coarse region of G6X4.

L218: This assertion seems to be mostly true. But I am struck by the oscillations in northern Alaska that are absent in the uniform resolution runs, and which coincide with the mesh transition zone. I think these are real errors. Similar errors are discussed in the context of the SURX4 grid in the following paragraph, but there is no mention of these oscillations in the other VR grids (albeit, they are less noisy than SURX4).

L270: Similarly, it looks to me that the vorticity field in 8a is rather oscillatory, especially in the tails of the vortices. I think the authors should investigate whether these are real errors, an artifact of the vorticity calculation, or something else. It would also be interesting to understand the sensitivity of these spurious structures (if they are indeed spurious) to the Smagorinsky coefficient.

L288: Can the authors provide the rationale for using different physics-dynamics-tracer coupling methods for hydrostatic vs. non-hydrostatic runs?

L307: "During its movement from the 2nd-refinement into the 1st-refinement region, the change in the grid size leads to little distortion on the tropical cyclone in each experiment." This sentence would be more substantiated if the authors provided a look at how the tropical cyclone fares as it crosses the transition, not just the final structure after it already passed the transition (e.g., Figure 3 in your Zarzycki et al 2013 citation).

L310: The minor disturbance described near where the cyclone was initiated is a common feature of dycores in DCMIP2016. Might be worth looking into whether this result has been published before.

L321: It's unclear to me what the first sentence of this paragraph referencing Ringler has to do with the rest of the paragraph. Could the authors clarify?

L324: "clone" should say "cyclone."

L348: More important than what? I'd suggest removing the "more" from the last sentence.

Conclusions: I would think that the larger errors found using the SUR generator is a notable conclusion of this paper.

Figure 4: In the caption "the quasi-uniform G7 and G8 cases" should probably say "G6 and G7 cases," since the l2 norms are defined w.r.t to G8, no?

---

## Author Comment (AC1) · 27 Sep 2020

**Formal reply to GMD-2020-150**

First, the authors would like to thank all the reviewers and editors for their assistance in the manuscript review process. Based on the comments of two anonymous Reviewers and our own consideration of improving this work, we have largely revised the initial version/preprint. The major changes are listed as follows. Detailed responses to each Reviewer are attached. We use the black font to indicate comments and questions by two Reviewers, and blue for our response. The italic font describes how the manuscript has been modified.

**Major revision**

1. Both reviewers suggested to give more details on the explicit diffusion option and to demonstrate its impact. As mentioned in the short reply to Reviewer#1, we have revised the configuration of explicit diffusion based on some further exploration and tests. In the initial preprint, the Smagorinsky diffusion with a mean length scale was used. In this version, the square of this mean length is replaced by the length product of two local crossing edges. This leads to several changes in the simulations: (i) the sensitivity to the Smagorinsky coefficient is reduced, now closer to the QU simulations; (ii) given the same Smagorinsky coefficient, the tropical cyclone magnitude increases as compared to the initial version, because the diffusion strength in the fine-resolution area is reduced. *More details have been elaborated in the main text (Section 2.3.1).* Smagorinsky is not used for tracer transport in this version, although the sensitivity due to this is limited. Meanwhile, a fourth-order hyperdiffusion for the horizontal velocity is activated. Its reference coefficient has been scaled. Activating this option is helpful in both pure dynamical core tests and moist physics tests. *The reasons have been elaborated in the revised manuscript (Section 2.3.2).*

2. *We have rerun all the experiments in this manuscript version based on this latest configuration.* The simulations are further improved due to the above mentioned modifications. The model code and running scripts have been updated as a reference. Thanks to the open interactive discussion of GMD, the preprint provides a basis for the discussion in this revision. The readers may also compare the results of this version with those in the preprint to clearly see the difference.

3. *We have presented a full-physics variable-resolution test.* This helps to examine the VR behavior under more complex nonlinear feedback. It mainly examines three issues:
   (i) using explicit diffusion (hyperdiffusion only, Smagorinsky + hyperdiffusion) only slightly diffuses the key physical object;
   (ii) using explicit diffusion can suppress generation of some highly unrealistic disturbances found in full-physics modeling, which are far away from the initial vortex (the refined region);
   (iii) providing a comparison of two grids: G6B3X16L4 and G6B3X16. G6B3X16 has more rapid resolution changes. This helps to examine whether a high-densification ratio may seriously harm the simulation.

4. In the initial version, the relative vorticity field in the main text is displayed on the raw triangular grid that defines vorticity. As mentioned, this will show some oscillations that actually reflect the mesh shape, not real errors. To avoid aliasing, *we have remapped the vorticity field to the Voronoi cell.* This remapping is used for both QU and VR results in the main text.

5. *We have thoroughly improved the language of this manuscript. Most of the main text has been reorganized and rewritten.*
   The revised manuscript and supplement will be uploaded in a few days. The current

structure of this manuscript is organized as follows.

1. Introduction

We have further emphasized the importance and value of VR modeling, more specifically, the multiresolution approach supported by an unstructured mesh model. Two major challenges of VR modeling have been further elaborated: to resolve the fine-scale fluid structures; to avoid adverse impacts due to mesh transition and the higher solution errors in the coarse-mesh region.

2. Model description

We have provided a broader introduction on the model framework, dynamics, physics packages. In particular, a detailed description about the choice and configuration of the explicit diffusion options (Smagorinsky and hyperdiffusion) is given.

3. Mesh generation

The presentation has been improved. Figures 1 and 2 are used for an illustration.

4. Dry atmosphere

The results are improved. Figures 3,4 focus on the fine-scale resolving ability, and the performance over the mesh transition zone. Figures 5,6,7 focus on the issue of solution errors, with a more detailed analysis. Figure 8 focus on multiregional VR modeling, and reveal the impact of activating the hyperdiffusion option.

5. Moist atmosphere

5.1 Simple physics

Figures 9,10,11 focus on the impact of varying the mesh-generation parameters. The results are improved and a more detailed analysis has been given. Figure 12 offers a comparison of two QU-VR groups, focusing on resolution sensitivity.

5.2 Full physics

Figures 13, 14 focus on the overall performance and the impacts of explicit diffusion. Figure 15 focuses on the impact of different mesh styles.

6. Summary

(i) The overall performance.
(ii) The impact of explicit diffusion.
(iii) The impact of the mesh styles.

Author note: The attached comments of two reviewers are copied from their PDF files. Some typos and format errors may exist during the transfer process.

Reviewer#1: Review of 'Configuration and Evaluation of a Global Unstructured Mesh Model based on the Variable-Resolution Approach'

In this manuscript, the variable-resolution (V-R) version of the GRIST model, based on Voronoi tessellations is described. The authors define the mesh generation process and explore multiple refinement approaches with a dry dynamical core test case (the Jablonowski-Williamson baroclinic wave) and a moist test case with simplified physics (the Reed-Jablonowski tropical cyclone). They subjectively (visually) and objectively (l2 errors, etc.) compare V-R simulations against quasi-uniform (Q-U) reference simulations and verify the V-R simulations perform generally as one would expect, particularly based on previous findings with V-R models using similar test cases.

V-R models have indeed been shown to be useful tools and multiple modeling centers are currently pursuing their development. Therefore, further evaluation and validation of such configurations is warranted, especially as V-R models become more commonly used for scientific research and application.

In general, the results here are a confirmation of robust performance rather than any overtly new physical insight. This makes GMD a suitable venue for such work. I do find the manuscript fairly underdeveloped, however. The model description is lacking, particular describing options specific to V-R dynamical cores such as scale-specific diffusion and model timestep. The simulations evaluating the ability of the tropical cyclone to move between resolutions are interesting but feel almost tacked on, with weak expansive discussion, particularly with regard to diffusion behavior. Some other useful and commonly-reported information is also omitted, such as computational scaling numbers.

While the manuscript wasn't illegible by any means, it did contain numerous grammatical errors that detract at times from the science.

I suggest major revisions. Again, this is more of an application of existing test cases to an existing model to essentially demonstrate that a V-R configuration is not performing poorly. For this to be a useful reference to other users of GRIST in the future, as well as a comparison benchmark for other modeling centers, some additional evaluation and breadth of discussion is warranted.

Major comments
• The model description in Section 2 is lacking.
– For example, it is unclear exactly what numerics are being applied. Finite volume, I assume? What is the vertical discretization? How close are the numerics to the Model for Prediction Across Scales (MPAS)?

Reply: To address this concern, *we have provided more details in the model description (Section 2.1)*. As mentioned earlier in the short reply, GRIST used some well-established techniques available in the icosahedral-/Voronoi-mesh modeling community, based on some publicly available documents. These details can be clearly found in the previous model development studies. *We have concisely summarized them in this revision.* GRIST is a different model and has its own unique aspects. Therefore, while MPAS has already examined the VR performance based on the centroidal Voronoi tessellation, it is still important to do our own exploration.

– It seems reasonable that the timestep of a global V-R simulation scales with the finest grid spacing to satisfy the CFL constraint, although this isn't explicitly stated. It would be helpful to note this, however, as some ill-posed V-R configurations can actually be more restrictive

from a stability perspective than their equivalent Q-U counterparts.

Reply: The current dycore timestep is mainly determined by the theoretical time step (e.g., we typically use an acoustic Courant number ~0.5 based on the smallest mean length scale, assume 350 m/s), although not all the tests use the maximum allowable step. The tracer transport and physics steps can be enlarged accordingly (e.g., DTP=1:5:10). The timestep for each test can be found in the supplement file.

We agree this comment very much: "some ill-posed V-R configurations can actually be more restrictive from a stability perspective…". *We have explicitly stated this issue in Section 2.3.* Actually, even for a QU model, a proper model configuration is also important, especially for high-resolution applications.

As mentioned, the fourth-order hyperdiffusion option of horizontal velocity has been activated. One of the reasons is that the Smagorinsky option needs a higher coefficient (as compared to uniform-mesh modeling) to suppress small-scale oscillations due to mesh transition in the VR mode, which in turn, restricts the numerical stability (especially the DTP splitting mode because diffusion is called at the step of physics). A background hyperdiffusion is more effective in suppressing these small-scale oscillations. The Smagorinsky scheme, even with a higher coefficient, can be inactive for certain regions. With hyperdiffusion, we can use a moderate Smagorinsky coefficient that does not challenge the stability, even in the tests using a highly variable mesh that may reach sub-10 km locally. *This has been added in the revised manuscript. (see Section 2.3.1, 2.3.2)*

– What is the vertical resolution of the model? Is this constant across all configurations, or correspondingly increased in either/both the V-R and higher-resolution Q-U runs? How does this compare to other models with published baroclinic wave and tropical cyclone test results?

Reply: Indeed, some studies suggested that the vertical resolution should increase with increasing horizontal resolution, but we have not considered the impact of vertical resolution. In all our tests, we use 30 full vertical levels that are basically identical to the CAM5 setup used by Reed and Jablonowski (2012; 10.1029/2011MS000099). This was also used in our earlier QU model tests. A similar 30-level setup was used by Gettelman et al. (2018; 10.1002/2017MS001227) and Zarzycki et al. (2014; 10.1175/MWR-D-13-00179.1) in their CAM-SE-VR modeling. Thus, the simulations in this work can be compared with these earlier studies given the same horizontal resolution. *We have mentioned the vertical resolution in the revision. (Section 2.1, the last paragraph)*

– Appealing aspects of V-R modeling are the computational savings when solving a regional problem. Do the authors have scaling numbers that could provide a more objective quantification of this? Should they expect the simulations to scale linearly with the number of degrees of freedom in the mesh? Is there additional overhead associated with refinement that causes this scaling to be sub-linear?

Reply: As mentioned, the time step is limited by the fine-resolution area (for the tests we have examined). Thus, compared to a fine-resolution uniform-mesh model, the VR model is definitely more economic as it reduces the total grid number. For the scaling issue, we have compared a pair of VR and QU grids (G6B3, G6B3X16L4, 368642 cells). These tests use the full-physics configuration described in the revised manuscript with the nonhydrostatic core. As shown in Figure 1 of this reply, the speed up ratios and parallel efficiency look similar (at least for this test). Note that some super-linear speedup ratios are found, we ascribe this to relatively inefficient cache access of indirect addressing when using a small number of cores. For more details about the parallel infrastructure, one may find in Liu et al. (2020, manuscript submitted to GMD).

[Figure]

Figure1 Tropical cyclone test with full physics: speed up ratio and parallel efficiency for quasi-uniform (qu) and variable-resolution (vr) runs (NDC, G6B3, 368642 cells), starting from 320 cores to 3840 cores. Computing environment: Intel CPU E5-2697V4, 2.6GHz, 32 cores, 128 GB/node. Note that this test used a different cluster from that used by Zhang et al. (2020), and the scaling performance cannot be strictly compared. It only gives a relative comparison of QU and VR.

• Along this line, it is unclear what (if any) modifications are made for the V-R configurations relative to the Q-U. A Smagorinsky diffusion is applied in the horizontal. Is there any additional scale-selective explicit diffusion such as hyperdiffusion, or does the flow-dependent Smagorinsky handle everything? The latter would imply a fairly diffusive scheme in an implicit sense.

Reply: As replied above, in the initial preprint, only a Smagorinsky diffusion is applied in the horizontal. Again, we emphasize that the Smagorinsky diffusion is indeed stronger if fully active, but not that diffusive. For VR modeling, a background hyperdiffusion is more helpful to suppress grid-scale oscillations due to increased mesh discontinuity. *The detailed reasons have been added to the manuscript (Section 2.3.1, Section 2.3.2).*

• The moist tropical cyclone test section is underdeveloped.

– A couple sentences of additional description are warranted. What is the surface configuration, what does the idealized moist physics consist of? Convection? Boundary layer parameterization? Surface fluxes? How else is the model initialized?

Reply: *This information has been added in the revision. (Section 2.2)*

– The cyclone moves through the mesh – how is this done? Is there a background flow or does the configuration rely on beta drift associated with gradients of Coriolis across the cyclone?

Reply: *We have added necessary information when describing this test case. (the 1st paragraph of Section 5.1)*

– I would postulate that relative vorticity would be a better quantity to evaluate when assessing potential distortion or wave reflection in a numerical accuracy sense (e.g., Figs. 7-8). Are there artifacts in this field during the TC transit?

Reply: *We have shown the relative vorticity field of a TC case in the supplement file. As you may see, there is no oscillation in this field. The vorticity field in this test is basically a mass of positive vorticity values, and does not show the fine-scale structure as in the baroclinic wave. So we do not include it in the main text.*

– Other relevant citations which could help contextualize the TC results with respect to dynamical core and diffusion are Zhao et al. [2012] and Reed et al. [2015].

Reply: *These two references have been introduced in the main text to demonstrate the*

*impact of model dynamics on the tropical cyclone simulations (the 2ⁿᵈ paragraph of Section5.1).*

• It is quite unclear exactly what the authors are showing in Fig. 12. Is the goal of this figure to show that V-R simulations are more sensitive to diffusion coefficient than a Q-U grid with the same setting(s)? In some ways, it is a natural finding that a cyclone transiting multiple grid spacing will 'feel' multiple diffusion scales, although as noted above, it isn't stated whether this diffusion explicitly scales with resolution or this is an implicit response. Further, in the abstract, the authors note that this 'suggest[s] the importance of parameter tuning,' although there is not enough description of the configuration to support this statement. Is this tuning just one 'number' for the whole mesh? I would recommend spending another paragraph or two explaining the importance of this finding in the context of the V-R validation exercise and how it pertains to the evaluated version of GRIST.

Reply: Thanks for this comment. As mentioned, the parametric sensitivity to the Smagorinsky coefficient has been largely reduced because a local-scale Smagorinsky eddy viscosity is used. The original formulation with a mean length scale implies stronger diffusion (than necessary) for the fine-resolution area. It also leads to higher stability restriction, especially when it comes to a high densification ratio (e.g., G6B3X16). *We have explicitly compared the scaled and unscaled formulation to demonstrate the impact of this scaling (see the last paragraph of Section 5.1 and the supplement file).*

• Is is unclear from the KE spectra how well the V-R runs are doing. For example, they could be accumulating spurious energy near the grid cell. It doesn't appear that they are from the spatial plots, however, the interpolation to the T106 Gaussian grid means nothing definitive can be said about the refined regions within the nests since those are below the truncation scale. There are a few ways to evaluate KE spectra within a regional model or regional patch, such as those proposed by Errico [1985] and Skamarock [2004]. I would recommend their exploration.

Reply: Thanks for this suggestion. *We have recomputed regional KE spectra (based on the new tests) over a selected regional domain using the discrete cosine transform (DCT) method. The related context has been modified.* The basic computational procedure has been given in the Appendix. *For KE spectra, we also performed additional tests to examine the impact of varying the reference hyperviscosity coefficient.*

Minor comments

• Lines 49-53. Wave reflection can be strongly influenced by other parameters than transition zone width, such as numerical method and grid staggering. See Ullrich and Jablonowski [2011].

Reply: *This information has been given in the introduction, used as a motivation.*

• It is unclear why both hydrostatic and non-hydrostatic cores are exercised here. Both test cases do not emphasize non-hydrostatic dynamics (being of relatively 'coarse' resolution compared to regional weather models), so it should be expected that both solutions look similar in the absence of some sort of erroneous formulation. There is nothing inherently wrong with testing both cores, although it is mentioned more frequently than probably necessary.

Reply: Indeed, the only reason of checking both cores is to verify their similarity and consistency at the hydrostatic regime. Such consistency is our expectation before performing these tests. Our earlier studies have confirmed this in the QU mode, so we expect that there is no abnormal behavior in the VR mode as well. Considering that this issue has been confirmed in the preprint, for the TC test, *only the nonhydrostatic core is experimented based on the latest configuration.* The readers may see the preprint for a reference.

• Section 2.2.2. is quite long, specific, and doesn't add a ton of 'added value' to the manuscript. I would recommend shortening this slightly; keeping the description of important parameters (e.g., $\gamma$, $\lambda$, etc.) and removing extraneous text.

Reply: Sorry for this. *We have reduced the content of this section (now Section 3.2).*

• Lines 192-193. Why is the iteration number different for these grid methods? Is there a quantitative reason, or was this a subjective design choice during mesh generation?

Reply: This is basically empirical. During the iteration, two criteria are used to stop the loop:

(i) reach a user-defined minimum iterative number;

(ii) the circumcenter of each triangle falls within its shape.

The formal check of criterion (ii) will only be activated after the minimum iterative number. In general, when more points are used, more iterative steps are required to meet the second criterion.

• I am not sure what this sentence means in the code and data availability section: 'GRIST is available at https://github.com/grist-dev, in private repositories. A way is provided for the editor and reviewers to access the code, which does not compromise their anonymity (to our best effort).' I would double-check that this all conforms to GMD's policies.

Reply: As mentioned earlier in the short reply, *we have modified some statements in the code and data part.* The code can be accessed publicly, while needs authorization.

• I believe both the baroclinic wave and tropical cyclone test case were part of the Dynamical Core Model Intercomparison Project (DCMIP) test suite. It may be worth reviewing multi-center reviews (such as Ullrich et al. [2017]) or references from other labs to see if there is any benefit in comparing results to those previously published using the same test cases.

Reply: *We have added this information (DCMIP) in the introduction. In the main text, we have provided the online link of DCMIP2016 to direct interested readers to this site.*

• Fig. 2. It is not 100% clear which (X4) mesh is being shown. Are all the V-R meshes so similar they functionally look like this? If they are not, the three different generator meshes should be plotted.

Reply: If one uses the same density function for mesh generation, the grids produced by different generators will look similar, but differ in detail. *We have replaced fig. 2 with the grids generated by three density functions used in this study.*

• Figs. 5, 8, and 9. Why do the black refinement isolines look 'jagged?' I assume the plotting software is struggling with cell areas right at a given threshold, would recommend smoothing for visualization.

Reply: Thanks for this suggestion. *We have smoothed the isolines of the cell size using the nearest neighboring average, with a repeating number 100.*

• Fig. 6. Recommend moving the reference slopes above the spectra so that they do not intersect the raw data.

Reply: Thanks for this suggestion. *Done.*

Typographical errors and grammar

As noted above, there are numerous – albeit generally minor – grammatical errors. This list is not meant to be exhaustive, but rather, a few obvious catches I noted while reading. I recommend a thorough proofread for grammar before resubmission.

• Line 37. ... while permitting...

• Line 43. ... while retaining or minimally degrading...

• Line 64. ... maintains tropical cyclones...

• Line 83. 'three difference initial point sets' is awkward phrasing.

• Line 95. ... developmental ... (?)

• Line 216. ... model level nearest to...

• Lines 232-233. 'Nevertheless...' sentence is awkward.

• Lines 252. Perhaps something like 'sign of the relative vorticity is flipped to account for hemispheric differences' or thereabouts.

    Reply: *We have rewritten most of the main text, and carefully improved the language.*

Reviewer#2: Review of Configuration and Evaluation of a Global Unstructured Mesh Model based on the Variable-Resolution Approach Zhou et al.

General Impressions

This study evaluates the performance of the variable-resolution configuration of a newer global model GRIST, and seeks to understand the various strengths and weaknesses of different refinement meshes. The authors provide results from both dry and moist idealized experiments that illustrate that the solution in the refined regions resemble the uniform high-resolution solutions. While this take home message is clear, I would like to see further analysis/discussion on why the errors tend to be larger in VR compared with the uniform resolution runs, examples that I point out specifically in the comments section, and also how the Smagorinsky operators are implemented in VR. After addressing these minor revisions, I think this manuscript is acceptable for publication in GMD.

Comments

L64: CAM has multiple dycores, each with distinct numerical properties, and so this statement can be misleading. I think the authors should consider mentioning that the Zarzycki study cited used the spectral-element dycore.

  Reply: *This information has been added in the revision (please see introduction).*

L88: This statement "[a] series of numerical tests was performed to examine the model reliability under more challenging conditions," reads like there are more challenging tests than the TC test-case, but the TC test-case is the most complex case used in this study.

  Reply: Thanks. *This statement has been revised.*

L108: If I recall correctly, the Smagorinsky coefficients scale with grid spacing. Is the density function used to determine the Smagorinsky coefficients?

  Reply: In this revised manuscript, we have used the product of local grid distances to replace the square of mean length scale in the Smagorinsky eddy viscosity. For the hyperdiffusion, its reference coefficient is scaled by the ratio of grid spacings. *The details have been given in Section 2.3.* When testing the scaled diffusion, we did experiment with the density function approach for evaluating the ratio, but this choice was not used in the production runs because we feel that this formulation relies on the theoretical relation between the density values and cell spacings.

Model and configurations: Can the authors include the number of vertical levels used in the simulations?

  Reply: There are 30 full vertical levels used in all the numerical tests of this work. *This has been mentioned in the revised paper. (the last paragraph of Section 2.1)*

L160: The authors argue that the densification ratio should be no larger than 1:4, and point to a citation that I can't seem to get access to. I'm having trouble interpreting this statement. Do the authors mean no less than 1:4? Would this then mean the refined grid spacing should be no less than a 1/4 of the coarser region grid spacing? If so, I can think of many spectral-element VR studies that use a much smaller ratio without having reported any serious errors. I could be misunderstanding entirely here, but I think this densification ratio and implications of some lower limit should be spelled out more clearly for the general reader.

  Reply: Liu and Yang (2017) is available at:

  https://doc.global-sci.org/uploads/Issue/CiCP/v5n21/521_1310.pdf

  Based on some MPAS-SW simulations of the 2D cosine bell advection and steady state shallow water flow (10242 and 40962 cells), they suggested that the densification ratio should better have a moderate value (e.g., 1:X, X<4). They showed that 1:2 or 1:3 overall generates smaller errors than 1:4. This was also shown in Ringler et al. (2011) that 1:4 indeed generates

greater errors than 1:2, given the same coarse-mesh resolution (their Figure 8).

The implications of these results, based on our understanding, are not to discourage one from using a higher densification ratio (e.g., X16). The practical impact of these increased solution errors (e.g., from 1:2 to 1:4) may not be serious enough to deteriorate the practical simulations. MPAS simulations with 1:16 ratio available in the literature do not report any serious problem as well.

We have also performed a comparison of two X16 grids (G6B3X16 and G6B3X16L4) based on the full-physics tropical cyclone test. The final results at day 10 are consistent (see Section 5.2), although two cyclones experience different mesh sizes during the movement. In general, we feel that when one considers to use a mesh with a relatively high densification ratio, having a more gradual and hierarchical way may potentially lead to better quality, but a single-high ratio like X16 is also acceptable. Of course, more practical modeling experience is required. *To avoid ambiguity, we have removed this statement in the revised paper.*

L197: The authors keep referring to grid imprinting in this paragraph. Am I to infer that they are only talking about the spurious waves being generated in the southern hemisphere, in the coarse region of the grid? These features seem to become less noisy when the coarse region increases its resolution, as one would expect. I think it should be stated that the coarser region of G5B3X4 is higher resolution than the coarse region of G6X4.

Reply: Thanks for this comment. Indeed, the imprinting in the southern hemisphere is mainly related to the coarser resolution and mesh irregularities. If one increases the resolution for the coarse part, these imprinting errors can be further reduced (i.e., convergence of the numerical errors can be guaranteed). *We have performed more detailed analysis on the solution errors in the revision (Figures 5-7).* Please see the related paragraphs.

L218: This assertion seems to be mostly true. But I am struck by the oscillations in northern Alaska that are absent in the uniform resolution runs, and which coincide with the mesh transition zone. I think these are real errors. Similar errors are discussed in the context of the SURX4 grid in the following paragraph, but there is no mention of these oscillations in the other VR grids (albeit, they are less noisy than SURX4).

Reply: Indeed, in the initial version, the oscillations in northern Alaska are not reflecting the grid shape, but real errors. They are caused by the increased mesh discontinuity in the VR mode (QU does not support this), and *have been well removed in this version because the hyperdiffusion option is now active.* The flow-dependent Smagorinsky option is inactive over certain regions. *The results now look better.*

L270: Similarly, it looks to me that the vorticity field in 8a is rather oscillatory, especially in the tails of the vortices. I think the authors should investigate whether these are real errors, an artifact of the vorticity calculation, or something else. It would also be interesting to understand the sensitivity of these spurious structures (if they are indeed spurious) to the Smagorinsky coefficient.

Reply: Similar to the last question, *this figure has also been improved due to the hyperdiffusion.* We also compared the results with or without hyperdiffusion (Figure 8). Note that the no-hyperdiffusion case shows more oscillatory solutions than that in the preprint, because the strength of the Smagorinsky diffusion has been reduced for the fine-mesh region in this version (due to the local length scale and a smaller coefficient). *We have added more discussion for this part.*

L288: Can the authors provide the rationale for using different physics-dynamics-tracer coupling methods for hydrostatic vs. non-hydrostatic runs?

Reply: In the initial manuscript, we used DTP split coupling for the nonhydrostatic runs

and non-split coupling for the hydrostatic runs simply to confirm that: (i) the nonhydrostatic solver behaves similarly to its hydrostatic counterpart under the hydrostatic regime; (ii) the DTP splitting does not degenerate the model performance when it is properly configured, as compared to a non-split version.

Our previous studies have confirmed these issues in the QU mode, so we hope there is no abnormal behavior in the VR mode as well. Running four combinations would be too much, so we only choose to use the mutually exclusive combinations. As this issue has been validated in the preprint, *only the nonhydrostatic solver with DTP splitting is experimented in the TC test of the revised manuscript.*

L307: "During its movement from the 2nd-refinement into the 1st-refinement region, the change in the grid size leads to little distortion on the tropical cyclone in each experiment." This sentence would be more substantiated if the authors provided a look at how the tropical cyclone fares as it crosses the transition, not just the final structure after it already passed the transition (e.g., Figure 3 in your Zarzycki et al 2013 citation).

Reply: Thanks for this suggestion. *We have provided more details regarding the movement of the tropical cyclone before day 10, in both simple physics and full physics tests.*

L310: The minor disturbance described near where the cyclone was initiated is a common feature of dycores in DCMIP2016. Might be worth looking into whether this result has been published before.

Reply: Thanks for sharing this point. *We have mentioned this in the revised paper.* For this case, the minor disturbance can be suppressed by explicit diffusion. The results of this version almost do not show this because the hyperdiffuion was activated. In the initial version, if we use a higher Smagorinsky coefficient, this minor disturbance can also be suppressed, but hyperdiffusion is more effective. This minor disturbance is not as unrealistic as the new case that we show in the full-physics test, because it is close to the movement path of the major tropical cyclone. Its sensitivity to explicit diffusion is also realistic.

L321: It's unclear to me what the first sentence of this paragraph referencing Ringler has to do with the rest of the paragraph. Could the authors clarify?

Reply: We apologize for your confusion. Here is the reason. For a VR model, the truncation errors are determined by the coarsest part, as Ringlet et al. (2011) and some other studies have pointed out. At a first glance, this seems to be a disadvantage for VR modeling because we cannot reduce the overall truncation errors when the local resolution increases. *As we have further emphasized in the introduction,* the purpose of increasing resolution is to resolve the meteorologically important fine-scale fluid structures. Thus, what we expect from a VR model is that the better resolved fluid structures are not adversely influenced by the greater errors caused by the coarse resolution (and other issues like wave distortion due to mesh transition). Our numerical tests have well supported this point, that is, the local fine-scale structure is more closely related to the fine-resolution, provided that the adverse impacts due to mesh transition and the coarse part can be well controlled. *We have improved this paragraph in the revised manuscript to avoid ambiguity.*

L324: "clone" should say "cyclone."

Reply: Thanks. *This has been corrected.*

L348: More important than what? I'd suggest removing the "more" from the last sentence.

Reply: Thanks. *We have rewritten the entire paragraph.*

Conclusions: I would think that the larger errors found using the SUR generator is a notable conclusion of this paper.

Reply: Thanks. *This statement has been added to the conclusion.*

Figure 4: In the caption "the quasi-uniform G7 and G8 cases" should probably say "G6 and G7 cases," since the l2 norms are defined w.r.t to G8, no?

Reply: The original statement was correct. Computing the top of this shaded area follows Fig. 10 of Jablonowski and Williamson (2006, 10.1256/qj.06.12). It requires using the highest and $2^{nd}$-highest resolution tests (G7 and G8 in our case) to compute the uncertainty of the reference solutions.

---

## Referee Report (RR1)

**Second review of 'Configuration and Evaluation of a Global Unstructured Mesh Model based on the Variable-Resolution Approach'**

I appreciate the updates to the manuscript the authors have made. It is greatly improved over the initial submission, both in terms of the scientific results as well as stylistic presentation. In particular, I think the model solutions look improved in the case of added explicit diffusion (in this case, hyperdiffusion), which also makes the simulations more comparable to other, more mature, variable-resolution (VR) approaches in literature. The more complete analysis of the tropical cyclone (TC) in mesh transition regions is also nice and verifies that the numerics perform adequately in the presence of a 'strong' forcing for weather and climate applications.

I have a few notes listed below that either should be addressed (or at least considered) by the authors, editor, and GMD proof staff before final publication. However, I consider these minor and they do not require additional model simulations or deep analysis (generally clarification, figure modification, etc.), so I recommend **publication in GMD following minor revisions**.

One note, the authors make reference to 'the initial preprint' quite a few times in addressing changes from the original GMDD submission. While I (and other reviewers) clearly understand what is meant here, it may be worth ensuring that GMD is OK with this presentation, as it will be referencing parts of the original submission not directly included in the final published manuscript that most will directly download/access.

**Comments on scientific results**

- Lines 83-84. '... can capture smoother cloud patterns and smoother mid-level jet structures...' this is a bit ambiguous since 'smoother' could also mean 'lower resolution.' I believe what the authors are trying to say here is that a continuous discretization in a unified VR model does not include explicit (artificial) lateral boundary condition discontinuities one would find in a traditional nested model living inside a global solution (e.g., WRF).
- Lines 156-157. These two sentences are a bit vague. I assume by 'generic and flexible' the authors imply that there is an interface/coupling layer that allows one to add parameterizations that are developed in isolation to the model (provided they return tendencies that match what is required by the dynamical core). If these two sentences are to be left in, I would add a bit more clarification.
- Sections 2.3.1 and 2.3.2. It is worth noting that since different forms of explicit diffusion are applied to both the uniform (UQ) and VR meshes for both Smagorinsky and hyperdiffusion likely lead to some of the differences seen in the comparisons of Figs. 5 and 7 [Jablonowski and Williamson, 2011]. Generally, for VR models diffusion is applied in a way such that the explicit diffusion in the cells in the coarse/fine regions match their UQ counterparts. Far from the transition, the design goal is for the local numerics of a QU/VR pair to be indistinguishable from one another (e.g., Zarzycki et al. [2014], Park et al. [2014]).

While it is not critical to the results here, it needs to be mentioned here that the different choices of diffusion between QU and VR means that a true apples-to-apples comparison is impossible. For example, I suspect one reason for the slightly counterintuitive results in Figs. 5 and 7 (i.e., lines 364-394) is due to addition diffusion in the VR runs, which will slightly increase error relative to an less diffused (and well-posed) UQ simulation of equivalent coarse resolution.

• Lines 380-388. A counterargument to applying a perturbation in both hemispheres is that the lack of specified baroclinic development in the southern hemisphere allows for large wave number errors

associated with grid imprinting to be more overtly realized in the test than it would be in a more complex simulation (e.g., wavenumber 4 imprinting associated with cubed-sphere corners or wavenumber 5 imprinting associated with pentagons in a hexagonal mesh, as in Fig. S3a).

• Lines 508-515. It is worth noting that the environment in the Reed and Jablonowski TC is conditionally unstable, and therefore some of this noise is almost inevitably going to arise as horizontal gravity waves, etc. induce resolved-scale overturning in the formerly quiescent atmosphere. So while turning up hyperdiffusion can suppress these 'far-field' gridpoint storms, it is unclear whether that is a truly desirable outcome. One could imagine a case where a developer tunes a diffusive operator to eliminate these instabilities, but this results in an overdiffusive configuration for real-world convection in the deep tropics.

**Considerations regarding figures and tables**

- I would actually move Tables S1 and S2 into the body of the main manuscript and reference accordingly. I referred to these particularly frequently when cross-validating the VR acronyms, and anything that requires more than a single reference probably should be in the primary manuscript. Perhaps Table 1 can be moved to supplemental since it is implicitly referenced in the last two columns of Tables S1/S2.
- There are continental outlines in most of the figures, although all simulations are done in the absense of land models (and without any zonal asymmetries). I would add a small disclaimer when describing Fig. 1 or Fig. 3 that all outlines are only showed for spatial reference and do not represent land masses/surface forcing in the simulations.
- I would zoom in on the TC in Fig. 13 as the results described in the text are nearly impossible to see.
- Fig. 8 has a small kink in the 28 km contour (towards the southeast quadrant). I actually assume this is due to the small natural imperfections in a spherical tessellation (e.g., pentagons) but it may be worth noting in the manuscript that that contour isn't in error.

**Typographical errors and grammar**

This list is not meant to be exhaustive, but rather, a few obvious catches I noted while reading.

- Line 85. 'Based on a VR configuration...'
- Line 110. '... coupling) are in Z20 and Z19.'
- Line 118. '... that is, using a staggered finite-volume...'
- Line 160. '... study, the suite of Reed and Jablonowski...'
- Line 174. Splitting is a relatively niche subject, would reference something like Gross et al. [2018] for readers that would like additional detail.
- Line 306. '... examine the ability to resolve fine-scale ...'
- Lines 336-337. 'For G5B3X4,...'
- Line 376. '... or Gaussian grid were predominant.'
- Line 469. '... looks smooth and...'
- Line 530. '... with more realistic weather...'

**References**

- M. Gross, H. Wan, P. J. Rasch, P. M. Caldwell, D. L. Williamson, D. Klocke, C. Jablonowski, D. R. Thatcher, N. Wood, M. Cullen, B. Beare, M. Willett, F. Lemarié, E. Blayo, S. Malardel, P. Termonia, A. Gassmann, P. H. Lauritzen, H. Johansen, C. M. Zarzycki, K. Sakaguchi, and R. Leung. Physics–Dynamics Coupling in Weather, Climate, and Earth System Models: Challenges and Recent Progress. *Mon. Weather Rev.*, 146(11):3505–3544, Nov 2018. ISSN 0027-0644. doi: 10.1175/MWR-D-17-0345.1.
- C. Jablonowski and D. L. Williamson. The pros and cons of diffusion, filters and fixers in atmospheric general circulation models. In *Numerical techniques for global atmospheric models*, pages 381–493. Springer, 2011.
- S.-H. Park, J. B. Klemp, and W. C. Skamarock. A comparison of mesh refinement in the global MPAS-A and WRF models using an idealized normal-mode baroclinic wave simulation. *Monthly Weather Review*, 142(10):3614–3634, 2014.
- C. M. Zarzycki, M. N. Levy, C. Jablonowski, J. R. Overfelt, M. A. Taylor, and P. A. Ullrich. Aquaplanet experiments using CAM's variable-resolution dynamical core. *Journal of Climate*, 27(14):5481–5503, 2014. doi: 10.1175/JCLI-D-14-00004.1.

---

## Author Response (AR2)

**Second review of 'Configuration and Evaluation of a Global Unstructured Mesh Model based on the Variable-Resolution Approach'**

I appreciate the updates to the manuscript the authors have made. It is greatly improved over the initial submission, both in terms of the scientific results as well as stylistic presentation. In particular, I think the model solutions look improved in the case of added explicit diffusion (in this case, hyperdiffusion), which also makes the simulations more comparable to other, more mature, variable-resolution (VR) approaches in literature. The more complete analysis of the tropical cyclone (TC) in mesh transition regions is also nice and verifies that the numerics perform adequately in the presence of a 'strong' forcing for weather and climate applications.

I have a few notes listed below that either should be addressed (or at least considered) by the authors, editor, and GMD proof staff before final publication. However, I consider these minor and they do not require additional model simulations or deep analysis (generally clarification, figure modification, etc.), so I recommend publication in GMD following minor revisions.

One note, the authors make reference to 'the initial preprint' quite a few times in addressing changes from the original GMDD submission. While I (and other reviewers) clearly understand what is meant here, it may be worth ensuring that GMD is OK with this presentation, as it will be referencing parts of the original submission not directly included in the final published manuscript that most will directly download/access.

Reply: We thank this reviewer for helpful comments. Considering that the preprint (and its supplement) can be directly downloaded from GMD's website, we keep this usage for the convenience of illustration. In the "code and data" section, we have explicitly stated that "*the preprint is available from the online link of this paper*". We have slightly revised the latest version. Detailed responses to the Reviewer are given as follows.

**Comments on scientific results**

• Lines 83-84. '... can capture smoother cloud patterns and smoother mid-level jet structures...' this is a bit ambiguous since 'smoother' could also mean 'lower resolution.' I believe what the authors are trying to say here is that a continuous discretization in a unified VR model does not include explicit (artificial) lateral boundary condition discontinuities one would find in a traditional nested model living inside a global solution (e.g., WRF).

Reply: Thanks for pointing out this. This sentence is modified as "*The VR model can capture smoother cloud patterns and smoother mid-level jet structures across the grid refined region...*".

• Lines 156-157. These two sentences are a bit vague. I assume by 'generic and flexible' the authors imply that there is an interface/coupling layer that allows one to add parameterizations that are developed in isolation to the model (provided they return tendencies that match what is required by the dynamical core). If these two sentences are to be left in, I would add a bit more clarification.

Reply: We have modified this paragraph:

"GRIST provides a general physics–dynamics coupling interface to incorporate various physics packages. A tailored package can be used as a plugin, and its development can benefit from the broad community resources. One may add a specific physics scheme to an existing physics package, or create an entirely new physics package as long as it is compatible with the current workflow and is scientifically reliable. The surface model (e.g., land or a mixed-layer ocean model), though not used in this study, is coupled in a point-to-point style, and can be shared by different physics configurations. Three physics packages are currently available as basis for continuous research and development. These packages are separate in the sense that they have different physics drivers and data structures. For completeness, we describe them in this section."

• Sections 2.3.1 and 2.3.2. It is worth noting that since different forms of explicit diffusion are applied to both the uniform (UQ) and VR meshes for both Smagorinsky and hyperdiffusion likely lead to some of the differences seen in the comparisons of Figs. 5 and 7 [Jablonowski and Williamson, 2011]. Generally, for VR models diffusion is applied in a way such that the explicit diffusion in the cells in the coarse/fine regions match their UQ counterparts. Far from the transition, the design goal is for the local numerics of a QU/VR pair to be indistinguishable from one another (e.g., Zarzycki et al. [2014], Park et al. [2014]).

While it is not critical to the results here, it needs to be mentioned here that the different choices of diffusion between QU and VR means that a true apples-to-apples comparison is impossible. For example, I suspect one reason for the slightly counterintuitive results in Figs. 5 and 7 (i.e., lines 364- 394) is due to addition diffusion in the VR runs, which will slightly increase error relative to an less diffused (and well-posed) UQ simulation of equivalent coarse resolution.

Reply: We agree that this is not "a true apples-to-apples comparison", but this is the best that we can do now. The comparison of error norms ensures that the VR configuration produces overall reasonable error behaviors: increasing the coarse-mesh resolution decreases the global error. Thus, the model can be used for more realistic weather and climate applications. The counterintuitive result (G8X4) is only found for the first few days. We will further think about this question and try to find more confirmative reasons. We have slightly modified the end of the 1st paragraph of Section 2.3.2 as:

"Thus, QU and VR models are applied by different forms of explicit diffusion in this manuscript. Future work may also need to examine the possible impact of hyperdiffusion in a QU model to better isolate its effect."

• Lines 380-388. A counterargument to applying a perturbation in both hemispheres is that the lack of specified baroclinic development in the southern hemisphere allows for large wave number errors associated with grid imprinting to be more overtly realized in the test than it would be in a more complex simulation (e.g., wavenumber 4 imprinting associated with cubed-sphere corners or wavenumber 5 imprinting associated with pentagons in a hexagonal mesh, as in Fig. S3a).

Reply: Indeed, the double-perturbation test makes the imprinting issue less conspicuous. For this part, what we are addressing is not "the imprinting issue does not exist", but that "its impact can be controlled". We believe that it is better to understand and control such impacts in a more realistic environment. In reality, the southern hemisphere is not silent. The consequence of this issue may not be as serious as in a steady state situation. The reduction in the relative error of G8X4 in relative to G7 (Fig. 7 v.s. Fig. 5b) supports this argument. This gives us confidence on the precision of the solver when used in a real-world application.

• Lines 508-515. It is worth noting that the environment in the Reed and Jablonowski TC is conditionally unstable, and therefore some of this noise is almost inevitably going to arise as horizontal gravity waves, etc. induce resolved-scale overturning in the formerly quiescent atmosphere. So while turning up hyperdiffusion can suppress these 'far-field' gridpoint storms, it is unclear whether that is a truly desirable outcome. One could imagine a case where a developer tunes a diffusive operator to eliminate these instabilities, but this results in an overdiffusive configuration for real-world convection in the deep tropics.

Reply: Thanks for pointing out this issue. Indeed, small disturbances are ubiquitous, and they may or may not excite new systems. For this particular case, it depends on the nonlinear feedback between dynamics and physics (for example, certain combination of physics schemes or the physics coupling itself may also remove such systems). We agree that it is not that straightforward to say what is right or wrong. We use the term "highly unrealistic", because we believe that the remote systems should not appear. But any way, the major purpose here is to show the role of explicit diffusion in a VR configuration under full physics-dynamics interaction.

We have slightly modified the first part of Section 5.1 as:

"The initial virtual temperature profiles are designed to be conditionally unstable in the troposphere. A small perturbation (either physical or computational) is more likely to excite new storms. Whether these additional signals are realistic depends on the situations."

**Considerations regarding figures and tables**

• I would actually move Tables S1 and S2 into the body of the main manuscript and reference accordingly. I referred to these particularly frequently when cross-validating the VR acronyms, and anything that requires more than a single reference probably should be in the primary manuscript. Perhaps Table 1 can be moved to supplemental since it is implicitly referenced in the last two columns of Tables S1/S2.

Reply: Tables S1, S2 now become Tables 1 and 2. Table 1 is Table S1 now.

• There are continental outlines in most of the figures, although all simulations are done in the absence of land models (and without any zonal asymmetries). I would add a small disclaimer when describing Fig. 1 or Fig. 3 that all outlines are only showed for spatial reference and do not represent land masses/surface forcing in the simulations.

Reply: We have added the disclaimer below Fig. 1 as: "*Throughout this paper, all land-sea outlines are only given for a spatial reference, and do not represent the geographical difference.*"

• I would zoom in on the TC in Fig. 13 as the results described in the text are nearly impossible to see.

Reply: This has been done for Fig. 13.

• Fig. 8 has a small kink in the 28 km contour (towards the southeast quadrant). I actually assume this is due to the small natural imperfections in a spherical tessellation (e.g., pentagons) but it may be worth noting in the manuscript that that contour isn't in error.

Reply: This information has been explicitly given in the caption of Fig. 8.

"Note that the small kink in the 28 km contour of (a) (c) simply reflects that the generated mesh occasionally has higher local irregularities at some areas."

**Typographical errors and grammar**

This list is not meant to be exhaustive, but rather, a few obvious catches I noted while reading.

• Line 85. 'Based on a VR configuration...'

• Line 110. '... coupling) are in Z20 and Z19.'

• Line 118. '... that is, using a staggered finite-volume...'

• Line 160. '... study, the suite of Reed and Jablonowski...'

• Line 174. Splitting is a relatively niche subject, would reference something like Gross et al. [2018] for readers that would like additional detail.

• Line 306. '... examine the ability to resolve fine-scale ...'

• Lines 336-337. 'For G5B3X4,...'

• Line 376. '... or Gaussian grid were predominant.'

• Line 469. '... looks smooth and...'

• Line 530. '... with more realistic weather...'

Reply: Thanks for these suggestions. We have corrected them in the main text. We also introduce Gross et al. (2018) to give a reference for splitting.

[revised manuscript text omitted]